# Distinct impact of IgG subclass on autoantibody pathogenicity in different IgG4-mediated diseases

Yanxia Bi[1,2], Jian Su[3,4], Shengru Zhou[5], Yingjie Zhao[1,2], Yan Zhang[1,2], Huihui Zhang[1,2], Mingdong Liu[1,2], Aiwu Zhou[2], Jianrong Xu[2], Meng Pan[5]*, Yiming Zhao[3,4]*, Fubin Li[1,2]*

[1]Center for Immune-Related Diseases at Shanghai Institute of Immunology, Ruijin Hospital, Shanghai Jiao Tong University School of Medicine, Shanghai, China; [2]Shanghai Institute of Immunology, Faculty of Basic Medicine; Key Laboratory of Cell Differentiation and Apoptosis of Chinese Ministry of Education, Shanghai Jiao Tong University School of Medicine, Shanghai, China; [3]Jiangsu Institute of Hematology, The First Affiliated Hospital of Soochow University, Suzhou, China; [4]Collaborative Innovation Center of Hematology, Soochow University, Suzhou, China; [5]Department of Dermatology, Rui Jin Hospital, Shanghai Jiao Tong University School of Medicine, Shanghai, China

**Abstract** IgG4 is the least potent human IgG subclass for the FcγR-mediated antibody effector function. Paradoxically, IgG4 is also the dominant IgG subclass of pathogenic autoantibodies in IgG4-mediated diseases. Here, we show that the IgG subclass and Fc-FcγR interaction have a distinct impact on the pathogenic function of autoantibodies in different IgG4-mediated diseases in mouse models. While IgG4 and its weak Fc-FcγR interaction have an ameliorative role in the pathogenicity of anti-ADAMTS13 autoantibodies isolated from thrombotic thrombocytopenic purpura (TTP) patients, they have an unexpected exacerbating effect on anti-Dsg1 autoantibody pathogenicity in pemphigus foliaceus (PF) models. Strikingly, a non-pathogenic anti-Dsg1 antibody variant optimized for FcγR-mediated effector function can attenuate the skin lesions induced by pathogenic anti-Dsg1 antibodies by promoting the clearance of dead keratinocytes. These studies suggest that IgG effector function contributes to the clearance of autoantibody-Ag complexes, which is harmful in TTP, but beneficial in PF and may provide new therapeutic opportunity.

## Editor's evaluation

The pathogenesis of human IgG4-mediated autoimmune diseases and the relevance of IgG4 autoantibodies remain incompletely understood and this study addresses an interesting issue on the role of IgG4 autoantibodies on disease pathology. By using FcgR knockout and engineered Abs, authors nicely show that IgG effector function mediates opposite functions, protecting or exacerbating the pathophysiology, depending on diseases.

## Introduction

IgG4, the least expressed IgG subclass in humans, is a highly relevant IgG subclass in a range of IgG4-mediated autoimmune diseases and IgG4-related diseases, both of which are still expanding (*Huijbers et al., 2018*; *Koneczny, 2018*; *Umehara et al., 2017*). It is generally accepted that IgG4 has the poorest effector function among all human natural IgG subclasses and is anti-inflammatory

*For correspondence:
pm10633@rjh.com.cn (MP);
zhaoyimingbox@163.com (YZ);
fubin.li@sjtu.edu.cn (FL)

(*Lighaam and Rispens, 2016*). This is due to the unique features of IgG4, including its low affinity to FcγRs and lack of capacity to activate complements, as well as its reduced ability to form immune complex due to a unique process referred to as Fab-arm exchange (*Lighaam and Rispens, 2016*; *Vidarsson et al., 2014*). The Fab-arm exchange between different IgG4 antibodies results in the formation of heterodimeric IgG4 molecules (i.e., bi-specific IgG4 antibodies) that only allow monovalent binding (*Koneczny, 2018*; *Lighaam and Rispens, 2016*). Given these features, it has been speculated that the choice of IgG4 subclass in autoantibodies in various relevant autoimmune diseases may be protective by reducing or preventing the pathogenic function of more harmful antibody classes or subclasses (*Rihet et al., 1992*). IgG4 autoantibodies targeting the acetylcholine receptor have been reported to protect monkeys from myasthenia gravis induced by matched IgG1 autoantibodies (*van der Neut Kolfschoten et al., 2007*). Allergen-specific IgG antibodies derived from hyperimmune beekeepers, mostly IgG4, have been reported to be protective in allergy patients (*Devey et al., 1989*) and mouse models (*Schumacher et al., 1996*). Phospholipase A$_2$-specific autoantibodies in patients with bee venom allergy switched from IgE to IgG4 after effective antigen-specific immunotherapy (*Akdis and Akdis, 2011*; *Devey et al., 1989*).

IgG4-mediated autoimmune diseases represent a unique category of more than 10 autoimmune conditions featured by the accumulation of pathogenic antigen-specific IgG4 autoantibodies. The pathogenic function of IgG4 autoantibodies has been either well established or highly suspected in the majority of these diseases (*Huijbers et al., 2018*; *Koneczny, 2018*). Among them, thrombotic thrombocytopenic purpura (TTP) and pemphigus foliaceus (PF) are two well-studied examples, in which both disease-triggering autoantibodies and their antigenic targets have been identified. In PF, autoantibodies bind to Dsg1 (desmoglein 1), a desmoglein protein highly expressed in the superficial layers of the epidermis and essential for adhesion between neighboring keratinocytes, and result in the loss of cell-cell adhesion and the blistering of skin (*Anhalt et al., 1982*; *Korman et al., 1989*; *Rock et al., 1989*). ADAMTS13 (a disintegrin and metalloproteinase with a thrombospondin type 1 motif, member 13), the critical enzyme maintaining the homeostasis of von Willebrand factor (vWF) in the plasma, is targeted by autoantibodies in acquired TTP (*Zheng, 2015*). In both PF and TTP patients, IgG4 is a major autoantibody subclass, and its levels correlate with disease activities (*Ferrari et al., 2009*; *Rock et al., 1989*; *Sinkovits et al., 2018*; *Warren et al., 2003*). Importantly, polyclonal anti-Dsg1 and ADAMTS13 autoantibodies enriched from patient plasma, as well as monoclonal anti-Dsg1 and -ADAMTS13 autoantibodies isolated from patients, are pathogenic in both in vitro assay and animal models (*Huijbers et al., 2018*; *Koneczny, 2018*). These studies, together with the study of other IgG4-mediated diseases and IgG4-related diseases (*Shiokawa et al., 2016*), have established that IgG4 autoantibodies are pathogenic and highly relevant to the development of these diseases (*Huijbers et al., 2018*; *Koneczny, 2018*).

It is, however, not clear whether the IgG4 subclass in these IgG4 autoantibodies has any impact on their pathogenicity. Most studies supported the notion that IgG4 is the most prevalent IgG subclass in anti-ADAMTS13 autoantibodies and is associated with disease relapse (*Ferrari et al., 2009*; *Sinkovits et al., 2018*). At the same time, IgG1 and IgG3 autoantibody levels appear to have a stronger correlation with disease severity in acquired TTP patients during the acute phase (*Bettoni et al., 2012*). In contrast, pathogenic anti-Dsg1 autoantibodies often have the IgG4 subclass, whereas non-pathogenic anti-Dsg1 autoantibodies often have the IgG1 subclass in endemic PF patients (*Aoki et al., 2015*; *Warren et al., 2003*). However, it is also reported that these pathogenic and non-pathogenic anti-Dsg1 autoantibodies have different binding epitopes (*Aoki et al., 2015*; *Li et al., 2003*). Therefore, despite that IgG4 autoantibodies are pathogenic and that some autoantibodies isolated from IgG4-mediated diseases (e.g., anti-Dsg1) are pathogenic in the form of scFv and Fab fragments (*Ishii et al., 2008*; *Rock et al., 1990*; *Yamagami et al., 2009*), it does not rule out the possibility that the choice of IgG4 in these autoantibodies is a protective mechanism against the otherwise more harmful antibody classes or subclasses.

Furthermore, it appears that antibodies with different modes of action, including effector antibodies, agonistic, and blocking antibodies, could be impacted by IgG subclasses and Fc-FcγR interactions in different ways. Both humans and mice have activating and inhibitory FcγRs that mediate or inhibit antibody effector functions, respectively. Either ablating activating FcγRs expression or ectopically overexpressing inhibitory FcγRIIB can protect autoimmune mice from premature mortality (*Clynes et al., 1998a*; *McGaha et al., 2005*), as well as arthritic antibody-induced joint inflammation

in murine models (*Ji et al., 2002*). Studies of blocking and agonistic antibodies, which exert their function in entirely different ways, also revealed a critical impact of both murine and human IgG subclasses and Fc-FcγR interactions on the activities of these antibodies (*Dahan et al., 2015*; *Liu et al., 2019*).

In this study, we reasoned that studying whether and how IgG subclasses and Fc-FcγR interactions impact autoantibody pathogenicity in the context of IgG4-mediated diseases can help us to understand the modes of action of IgG4 autoantibodies and disease pathogenesis. Anti-Dsg1 and ADAMTS13 autoantibodies isolated from IgG4-mediated diseases were investigated in physiologically relevant models where intact human IgG autoantibodies can interact with both their antigenic targets and Fc-receptor expressing cells.

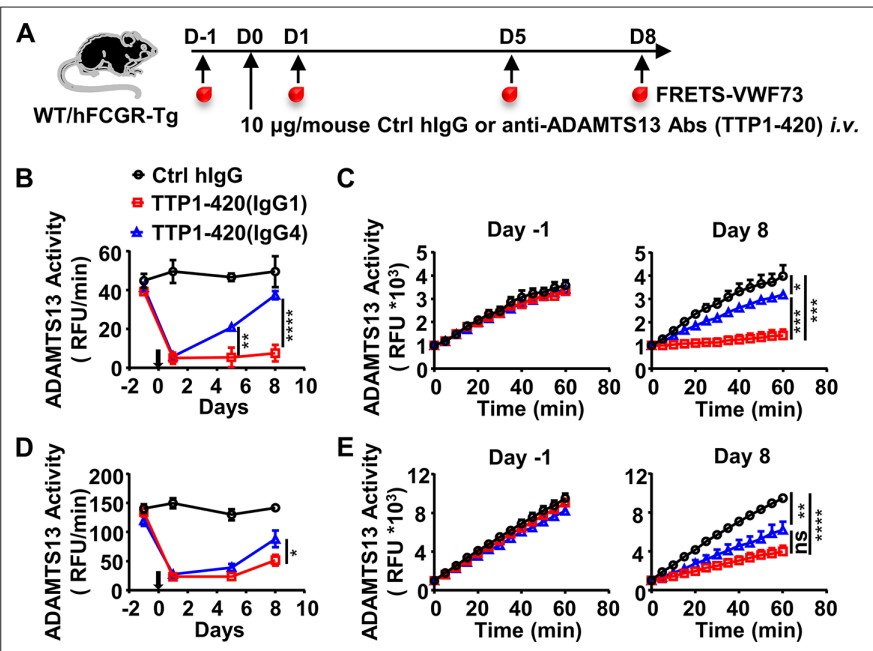

**Figure 1.** IgG4 is less pathogenic than IgG1 in anti-ADAMTS13 autoantibodies. (**A**) Schematic diagram of the experimental design. In brief, wild-type (WT) C57BL/6 or hFCGR-Tg mice were treated with 10 µg of control human IgG (Ctrl hIgG, n≥3), or anti-ADAMTS13 TTP1-420(IgG1) or TTP1-420(IgG4) (n≥4) on day 0 through tail vein injection, blood was collected on day −1, day 1, day 5, and day 8 and analyzed for ADAMTS13 activity. (**B–E**) Plots showing ADAMTS13 activity in the plasma of WT (**B, C**) or hFCGR-Tg (**D, E**) mice treated with the indicated antibodies at the indicated time points and analyzed by the FRETS-VWF73 assay, presented as relative fluorescence units (RFU) changing rates over time (RFU/min) (**B, D**), and the RFU change within 1 hr on day −1 and day 8 (**C, E**). Mean ± SEM values are plotted. Two-way ANOVA with Tukey's (**B, D**) or Sidak's (**C, E**) multiple comparisons tests. *p<0.05, **p<0.01, ***p<0.001, ****p<0.0001. A representative of two independent experiments is shown.

The online version of this article includes the following source data and figure supplement(s) for figure 1:

**Source data 1.** Related to *Figure 1B*.

**Source data 2.** Related to *Figure 1C*.

**Source data 3.** Related to *Figure 1C*.

**Source data 4.** Related to *Figure 1D*.

**Source data 5.** Related to *Figure 1E*.

**Source data 6.** Related to *Figure 1E*.

**Figure supplement 1.** Binding kinetics of anti-ADAMTS13 IgG antibodies to ADAMTS13 and mouse FcγRs.

**Figure supplement 2.** The decrease and recovery of plasma ADAMTS13 activity in TTP1-420(IgG1)-treated hFCGR-Tg mice.

## Results

### IgG4 is less pathogenic than IgG1 in anti-ADAMTS13 autoantibodies

To directly test whether the IgG subclass of anti-ADAMTS13 autoantibodies has any impact on their pathogenicity, a pathogenic monoclonal anti-ADAMTS13 antibody (clone TTP1-420) previously isolated from TTP patients and confirmed in mice (*Ostertag et al., 2016a*; *Ostertag et al., 2016b*) was expressed as either human IgG4 or IgG1 antibodies. These antibodies were confirmed to have similar binding kinetics to human ADAMTS13 (*Figure 1—figure supplement 1A*). Anti-ADAMTS13 autoantibodies have been previously shown to mediate their pathogenic effect by inhibiting the enzymatic activity of ADAMTS13, which leads to an increased ultra-large form of vWF and platelet binding and activation (*Zheng, 2015*). To evaluate the pathogenicity of TTP1-420(IgG4) and TTP1-420(IgG1) autoantibodies, their impact on ADAMTS13 enzymatic activity was analyzed first in WT mice (*Figure 1A*). As shown in *Figure 1B*, both TTP1-420(IgG4) and TTP1-420(IgG1)-treated WT mice displayed a significant reduction in ADAMTS13 activity soon after the treatment. However, the recovery of ADAMTS13 activity is significantly faster in TTP1-420(IgG4)-treated mice than in TTP1-420(IgG1)-treated mice, with almost complete recovery by day 8 in the former group and nearly no recovery in the latter group (*Figure 1B and C*).

To further evaluate the pathogenicity of TTP1-420(IgG4) and TTP1-420(IgG1) autoantibodies in the more physiologically relevant model with human Fcγ-receptor background, FcγR-humanized mice (hFCGR-Tg) that recapitulate the expression profile of human FcγRs (*Smith et al., 2012*) were used (*Figure 1A*). Consistently, TTP1-420(IgG4) autoantibodies also displayed weaker pathogenicity as compared to TTP1-420(IgG1) autoantibodies in FcγR-humanized mice (*Figure 1D, E*). These results suggest that while both IgG4 and IgG1 anti-ADAMTS13 autoantibodies can inhibit ADAMTS13 activity, IgG4 autoantibodies are less pathogenic as compared to IgG1 autoantibodies. Therefore, it appears that switching to the IgG4 subclass in anti-ADAMTS13 autoantibodies represents a protective mechanism. The ADAMTS13 activity in hFCGR-Tg mice treated with the same amount of TTP1-420(IgG1) was later recovered (*Figure 1—figure supplement 2A-C*), suggesting our model represents an acute model for evaluating the pathogenicity of anti-ADAMTS13 antibodies before the recovery of ADAMTS13 activity.

### Activating FcγR-mediated IgG effector function enhances the pathogenicity of anti-ADAMTS13 autoantibodies

Human IgG4 and IgG1 are very different in mediating FcγR-dependent effector function, with IgG4 being much less potent due to its weaker binding affinity to activating FcγRs (*Bruhns et al., 2009*; *Vidarsson et al., 2014*), including both mouse and human activating FcγRs (*Bruhns et al., 2009*; *Derebe et al., 2018*; *Liu et al., 2019* and *Figure 1—figure supplement 1B, C*, and *Supplementary file 1a*). To understand the basis of the differential pathogenicity of TTP1-420(IgG4) and TTP1-420(IgG1) autoantibodies, we investigated whether Fc-FcγR interaction impacts on anti-ADAMTS13 autoantibody pathogenicity using FcγR-deficient mice (FcγRα-null) (*Figure 2A*). As shown in *Figure 2B*, while TTP1-420(IgG1)-treated WT and FcγRα-null mice displayed comparable levels of reduction in ADAMTS13 activity at the early time points (day 1 and day 5) (*Figure 2B*), the recovery of ADAMTS13 activity is significantly faster in FcγRα-null mice (*Figure 2B, C*).

These results suggest that while FcγR-mediated function is not essential for the pathogenicity of anti-ADAMTS13 autoantibodies, it has an enhancing effect. Further investigation in Fc receptor common γ-chain deficient mice (*Fcer1g*$^{-/-}$), which lack functional activating FcγRs, showed that *Fcer1g*$^{-/-}$ mice have accelerated recovery of ADAMTS13 enzyme activity (*Figure 2A, D, E*), suggesting that activating Fcγ receptors are responsible for the enhancing effect of FcγRs to the pathogenicity of anti-ADAMTS13 autoantibodies.

To rule out the possibility that non-FcγR factors are responsible for the reduced pathogenic function of anti-ADAMTS13 autoantibodies in FcγRα-null and *Fcer1g*$^{-/-}$ mice, WT mice were treated with two variants of TTP1-420(IgG1) autoantibodies with different FcγR-binding properties (*Figure 2A*, *Supplementary file 1a and b*): (1) the N297A variant with reduced FcγR-binding and effector function (*Figure 1—figure supplement 1B, C* and references *Sazinsky et al., 2008*); (2) the GASDALIE variant with enhanced FcγR-binding and effector function (*Figure 1—figure supplement 1B, C* and *Bournazos et al., 2019*). As shown in *Figure 2F*, while both TTP1-420(N297A) and TTP1-420(GASDALIE) autoantibodies induced a significant reduction in ADAMTS13 enzyme activity

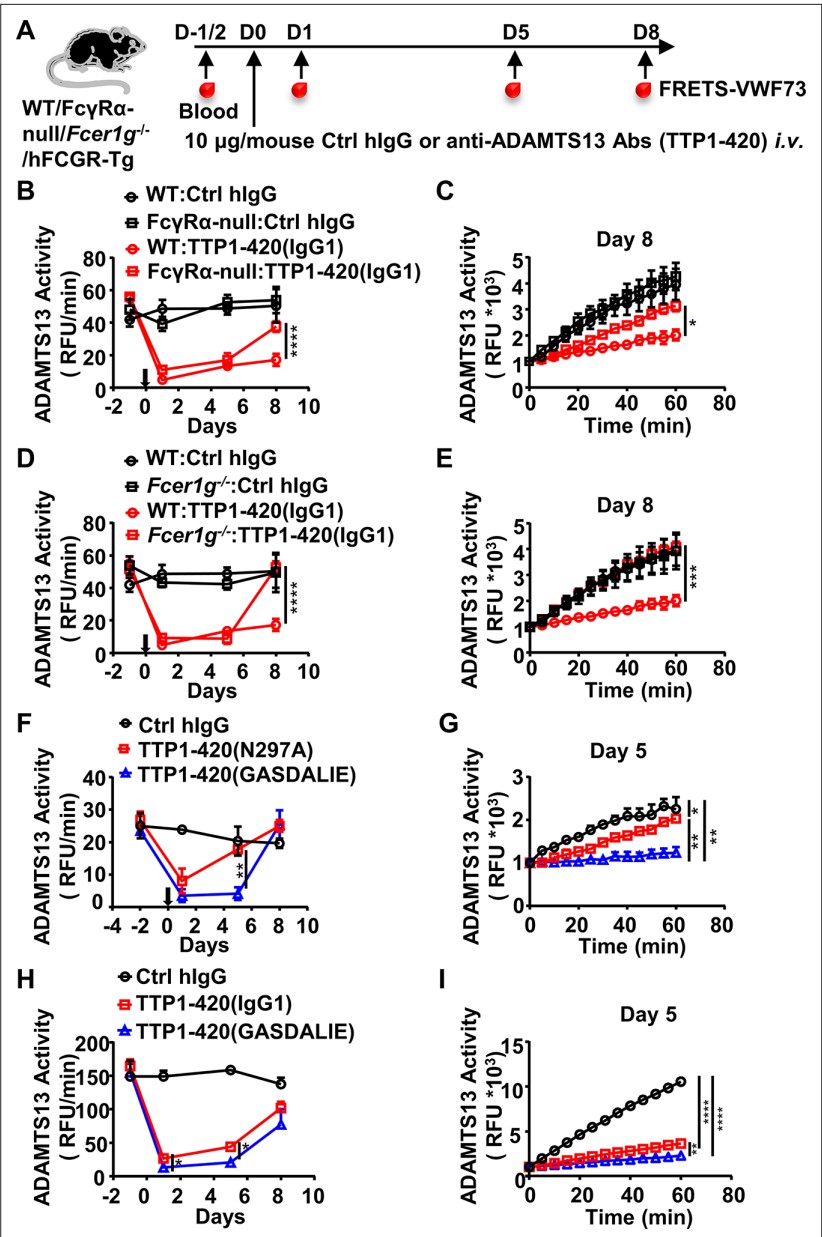

**Figure 2.** The protective effect of the IgG4 subclass in anti-ADAMTS13 autoantibodies is due to reduced FcγR-mediated antibody effector function. (**A**) Schematic diagram of the experimental design. In brief, WT, FcγRα-null, *Fcer1g⁻/⁻*, or hFCGR-Tg mice (Ctrl hIgG, n=3~5; anti-ADAMTS13, n=4~5) were treated and analyzed as in *Figure 1A*. (**B–E**) Plots showing ADAMTS13 activity in the plasma of WT and FcγRα-null mice (**B, C**), WT and *Fcer1g⁻/⁻* mice (**D, E**), WT mice (**F, G**) or hFCGR-Tg mice (**H, I**) treated with the indicated antibodies and analyzed at indicated time points as in *Figure 1B and C*, presented as relative fluorescence units (RFU) changing rates over time (RFU/min) (**B, D, F, H**), and the RFU change within 1 hr on the indicated days (**C, E, G, I**). Mean ± SEM values are plotted. Two-way ANOVA with Sidak's (**B–E, G, and I**) or Tukey's (**F and H**) multiple comparisons tests. *p<0.05, **p<0.01, ***p<0.001, ****p<0.0001. A representative of two independent experiments is shown.

The online version of this article includes the following source data for figure 2:

**Source data 1.** Related to *Figure 2B*.

**Source data 2.** Related to *Figure 2C*.

**Source data 3.** Related to *Figure 2D*.

**Source data 4.** Related to *Figure 2E*.

**Source data 5.** Related to *Figure 2F*.

*Figure 2 continued on next page*

*Figure 2 continued*

**Source data 6.** Related to *Figure 2G*.

**Source data 7.** Related to *Figure 2H*.

**Source data 8.** Related to *Figure 2I*.

(*Figure 2F*), TTP1-420(N297A)-treated mice recovered much faster (*Figure 2F, G*). The pathogenicity of non-mutated IgG1 and GASDALIE variant was also compared side by side in hFCGR-Tg mice. Consistently, both TTP1-420(IgG1) and TTP1-420(GASDALIE)-treated hFCGR-Tg mice induced a significant reduction in ADAMTS13 activity soon after the treatment (*Figure 2H*), and the recovery of ADAMTS13 activity is faster in TTP1-420(IgG1)-treated mice than in TTP1-420(GASDALIE)-treated mice (day 1 and day 5) (*Figure 2H, I*).

These results suggest that while the activating FcγR-mediated effector function is not absolutely required for the pathogenic function of anti-ADAMTS13 autoantibodies, it does have a critical enhancing effect, likely by depleting ADAMTS13-autoantibodies immune complex.

## IgG4 autoantibodies deplete relatively less ADAMTS13 in acquired TTP patients

To investigate whether IgG4 autoantibodies have a less depleting effect on ADAMTS13, plasma samples of a cohort of 44 acquired TTP patients were analyzed (*Supplementary file 1c*). The majority of these samples were confirmed to contain anti-ADAMTS13 IgG autoantibodies, with 43 out of 44 TTP samples having anti-ADAMTS13 autoantibody signal levels that are twofold higher than healthy control (HC) average levels, which were considered as the background (*Figure 3A, F*). All TTP samples were confirmed to have severely reduced ADAMTS13 activity (with all samples having less than 5% ADAMTS13 activity) (*Figure 3B*). Consistent with previous reports (*Bettoni et al., 2012*; *Ferrari et al., 2009*), ADAMTS13 antigen levels were also reduced (*Figure 3C*), and a correlation between ADAMTS13 activity and antigen levels was observed (*Figure 3D*). Further analysis of IgG1 and IgG4 anti-ADAMTS13 autoantibodies showed that most TTP samples have both IgG1 and IgG4 autoantibodies (*Figure 3E, F*) with an inverse correlation (*Figure 3F*).

Interestingly, while IgG1 autoantibodies have a significant inverse correlation with ADAMTS13 antigen and activity levels, IgG4 autoantibodies do not (*Figure 3G-J*), suggesting that IgG4 autoantibodies have relatively less impact on ADAMTS13 antigen and activity levels.

Given the inverse correlation between IgG1 and IgG4 autoantibodies in these TTP samples, they were divided into two groups: IgG1- and IgG4-dominant TTP samples, respectively (*Figure 3K*). Notably, while these two groups have similar ADAMTS13 activity levels (*Figure 3L*), the IgG4-dominant group has significantly higher ADAMTS13 antigen levels (*Figure 3M*). These results suggest that IgG4 anti-ADAMTS13 autoantibodies confer relatively less ADAMTS13 depletion.

## IgG4 and IgG1 anti-Dsg1 autoantibodies have shared binding epitopes and different abundance in PF patients

Previously, it has been described that the IgG1 anti-Dsg1 autoantibodies are observed in both healthy subjects and endemic PF patients, and the accumulation of IgG4 anti-Dsg1 autoantibodies is a key step in the development of the disease (*Warren et al., 2003*). At the same time, the binding epitope of anti-Dsg1 autoantibodies is critical for the pathogenicity of anti-Dsg1 autoantibodies (*Li et al., 2003*), raising the possibility that IgG4 and IgG1 anti-Dsg1 autoantibodies bind to different Dsg1 epitopes (*Aoki et al., 2015*). To investigate whether the binding epitope represents a major difference between IgG1 and IgG4 anti-Dsg1 autoantibodies, serum samples of a cohort of 53 PF patients admitted to Rui Jin Hospital, Shanghai, China, were analyzed for the total levels and subclasses of IgG anti-Dsg1 autoantibodies (*Supplementary file 1d*). Consistent with previous studies (*Warren et al., 2003*), PF patients have both IgG4 and IgG1 anti-Dsg1 autoantibodies (*Figure 4A* and *Figure 4— figure supplement 1A*). However, IgG4 autoantibodies are much more abundant than IgG1 autoantibodies (*Figure 4A*). Based on the clinical symptoms and medication, these PF patients were assigned to the 'stable' and 'active' groups. Both groups have more IgG4 autoantibodies than IgG1 autoantibodies (*Figure 4—figure supplement 1A*). The 'active' PF samples have higher autoantibody

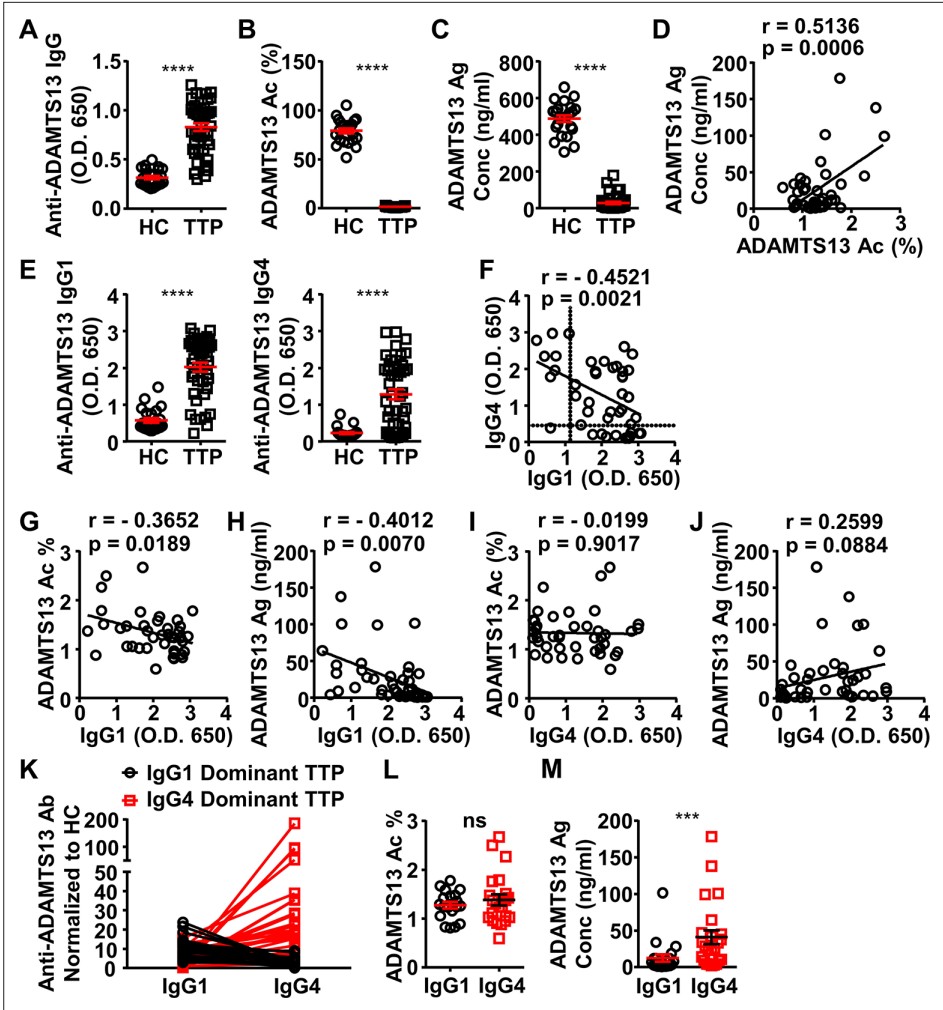

**Figure 3.** ADAMTS13-specific IgG1 levels in the plasma of acquired thrombotic thrombocytopenic purpura (TTP) patients inversely correlate to the ADAMTS13 Ag levels and activity. (**A–C**) Plots showing ADAMTS13-specific IgG (**A**), ADAMTS13 activity (**B**), and ADAMTS13 antigen (**C**) levels in the plasma of healthy control (HC) (n=23) and acquired TTP patients during the acute phase (n=44). (**D**) Plot showing correlation analysis of ADAMTS13 activity with ADAMTS13 Ag concentration in TTP patients. (**E**) ADAMTS13-specific IgG1 (left panel) and IgG4 (right panel) levels in the plasma of TTP patients and HC. (**F**) Plot showing correlation analysis of ADAMTS13-specific IgG1 with IgG4 levels in TTP patients, with the threshold for IgG1 and IgG4 autoantibodies (two times of HC average values) annotated. (**G–J**) Plots showing correlation analysis between the anti-ADAMTS13 IgG1 (**G, H**) and IgG4 (**I, J**) levels with ADAMTS13 activity (**G, I**) and ADAMTS13 Ag (**H, J**) levels in TTP patients, respectively. (**K**) Plots showing IgG1 and IgG4 anti-ADAMTS13 levels in TTP plasma samples normalized to HC, with IgG1-dominant TTP and IgG4-dominant TTP samples annotated. (**L, M**) Plot showing ADAMTS13 activity (**L**) and antigen (**M**) levels in TTP plasma samples as annotated in (**K**). Each symbol is derived from an individual plasma sample. Mean ± SEM values are plotted. Unpaired nonparametric Mann-Whitney test (**A, B, C, E, L, M**) or linear regression analysis (**D, F–J**). ***p<0.001, ****p<0.0001, ns, non-significant.

The online version of this article includes the following source data for figure 3:

**Source data 1.** Related to *Figure 3A*.

**Source data 2.** Related to *Figure 3B*.

**Source data 3.** Related to *Figure 3C*.

**Source data 4.** Related to *Figure 3D*.

**Source data 5.** Related to *Figure 3E*.

**Source data 6.** Related to *Figure 3E*.

**Source data 7.** Related to *Figure 3F*.

*Figure 3 continued on next page*

*Figure 3 continued*

**Source data 8.** Related to *Figure 3G*.

**Source data 9.** Related to *Figure 3H*.

**Source data 10.** Related to *Figure 3I*.

**Source data 11.** Related to *Figure 3J*.

**Source data 12.** Related to *Figure 3K*.

**Source data 13.** Related to *Figure 3L*.

**Source data 14.** Related to *Figure 3M*.

levels as compared to the 'stable' PF samples, regardless of what IgG subclasses were considered (*Figure 4—figure supplement 1A*).

The binding epitopes of IgG4 and IgG1 anti-Dsg1 autoantibodies were analyzed based on their binding to a series of Dsg1/Dsg2 chimeric proteins containing different Dsg1 extracellular domains (*Figure 4—figure supplement 1B*). Interestingly, all the tested samples with significant IgG1 antibodies showed that their IgG1 and IgG4 anti-Dsg1 autoantibodies have the same chimeric antigen-binding profiles (*Figure 4B* and *Figure 4—figure supplement 1C*). These results suggest that IgG1 and IgG4 anti-Dsg1 autoantibodies have different abundance but shared binding epitopes, at least in the patient samples analyzed in our study, and that the IgG subclass of these anti-Dsg1 autoantibodies could be a key variable in PF pathogenesis.

## IgG4 is not less, if not more, pathogenic than IgG1 in anti-Dsg1 autoantibodies

To investigate the impact of IgG4 subclass on the pathogenicity of anti-Dsg1 autoantibodies, we studied anti-Dsg1 autoantibodies previously isolated from PF patients and proven to be pathogenic in both neonatal mouse and human tissue-culture models (*Ishii et al., 2008*; *Yamagami et al., 2009*). Two anti-Dsg1 antibody clones (PF1-8-15 and PF24-9) were produced as IgG1 and IgG4 antibodies and confirmed to bind to Dsg1 with similar kinetics (*Figure 4—figure supplement 2A*). When tested in neonatal mice, both human IgG4 and IgG1 anti-Dsg1 antibodies induced epidermal acantholysis (*Figure 4—figure supplement 2B, C*). We also confirmed that the Fab fragment of anti-Dsg1 antibodies was sufficient to induce blisters in neonatal mice using papain-digested IgG1(PF24-9), at comparable levels as intact IgG1 and IgG4 antibodies (*Figure 4—figure supplement 2B, C*).

Since pemphigus is a chronic disease that often affects adults and older adults in the context of ongoing immune responses (*Schmidt et al., 2019*), and the neonatal mouse model is limited by the short time (usually for a few hours) allowed for investigation and its underdeveloped immune system, we developed a PF model based on adult mice. This model allows for the study of a complete pathological response once the autoantibodies are provided, including both the onset of blistering and the resolving of skin lesions. As shown in *Figure 4—figure supplement 3A*, adult hFCGR-Tg mice treated with PF24-9(IgG1) autoantibodies could reproduce the clinical, histological, and immunological features of a skin lesion in PF patients, including erosions that heal with crusting and scaling, intercellular deposition of human IgG in the epidermis at prelesion and lesion sites, and histopathological changes including ear and skin epidermal blistering formation. In addition, we observed the thickened epidermis and leukocyte infiltration at the lesion sites during the disease onset, as well as crusting and scaling during the recovery (*Figure 4—figure supplement 3A, B*), which generally last for a few days depending on the amount of autoantibody administered. These observations are consistent with previous analyses of skin samples of pemphigus patients (*Furtado, 1959*; *Radoš, 2011*) and the recent description of lymphocyte infiltration at the lesion sites in patients (*Yuan et al., 2017*; *Zhou et al., 2019*). Therefore, the adoptive transfer of anti-Dsg1 antibodies is sufficient to initiate pemphigus in adult mice, in which the pathogenicity of anti-Dsg1 autoantibodies can be studied in the process of disease onset and resolution and in the context of an intact immune system.

To investigate whether IgG subclass impacts on the pathogenicity of anti-Dsg1 autoantibodies, the pathogenicity of high dosage of PF24-9(IgG4) and PF24-9(IgG1) autoantibodies were evaluated in hFCGR-Tg mice (*Figure 4—figure supplement 3C*). When skin and ear tissues were analyzed, epidermal blistering and leukocyte infiltration were observed in both PF24-9(IgG4) and PF24-9(IgG1) autoantibody-treated mice (*Figure 4—figure supplement 3D*). After treatment, we detected

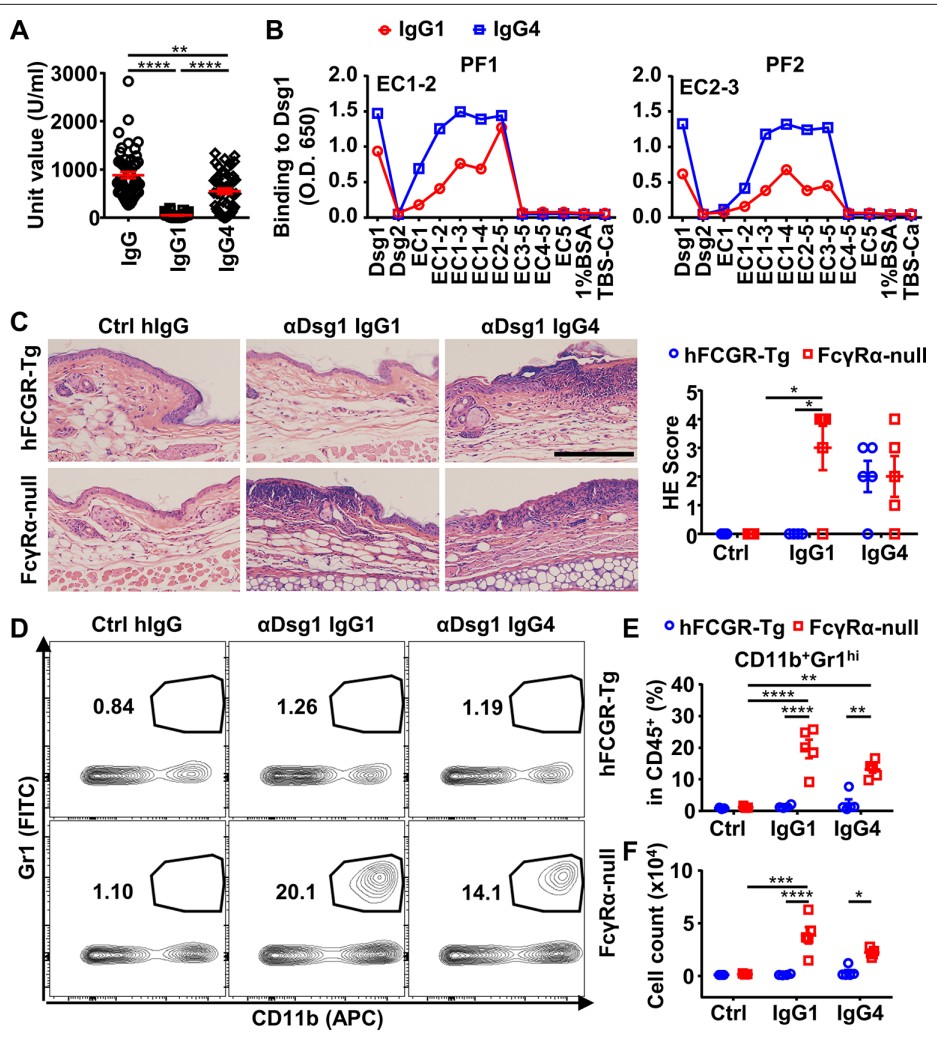

**Figure 4.** IgG4 is not less, if not more, pathogenic than IgG1 in Dsg1 autoantibodies, and both could be exacerbated by FcγRs deficiency. (**A**) Plots showing the unit values of indicated Dsg1-specific antibodies in the serum of pemphigus foliaceus (PF) patients (n=53). (**B**) Plots showing the levels of IgG1 and IgG4 antibodies in two PF patients that bind to Dsg1, Dsg2, or Dsg1/Dsg2 chimeric molecules containing the indicated Dsg1 EC domains. (**C**) Representative photos and scores showing the hematoxylin-eosin (HE) staining results of ears of hFCGR-Tg and FcγRα-null mice 3 days after being treated with 0.4 mg of Ctrl hIgG (n=3), or PF24-9(IgG1) or PF24-9(IgG4) (n≥4) through tail vein injection. Scale bars: 200 μm. (**D–F**) Representative flow cytometry profiles (**D**) and plots showing the percentage (**E**) and cell number (**F**) of infiltrating neutrophils (CD11b⁺Gr1ʰⁱ) among leukocytes (CD45⁺) in the ears of mice in (**C**). Each symbol is derived from an individual PF patient (**A**) or an individual mouse (**C, E, F**). Mean ± SEM values are plotted (**A, C, E, F**). One-way ANOVA (**A**) and two-way ANOVA (**C, E, F**) with Tukey's multiple comparisons test. *p<0.05, **p<0.01, ***p<0.001, ****p<0.0001. A representative of two independent experiments is shown.

The online version of this article includes the following source data and figure supplement(s) for figure 4:

**Source data 1.** Related to *Figure 4A*.

**Source data 2.** Related to *Figure 4B*.

**Source data 3.** Related to *Figure 4B*.

**Source data 4.** Related to *Figure 4C*.

**Source data 5.** Related to *Figure 4E*.

**Source data 6.** Related to *Figure 4F*.

**Figure supplement 1.** IgG4 and IgG1 anti-Dsg1 autoantibodies have shared binding epitopes and different abundance in pemphigus foliaceus (PF) patients.

*Figure 4 continued on next page*

*Figure 4 continued*

**Figure supplement 2.** Properties of anti-Dsg1 antibodies.

**Figure supplement 3.** An adult mouse model of pemphigus foliaceus.

**Figure supplement 4.** Gating strategy used for CD11b⁺ myeloid cells and CD11b⁺Gr1ʰⁱ neutrophils.

**Figure supplement 5.** IgG1 and IgG4 anti-Dsg1 antibodies induce more myeloid cell infiltration in the absence of FcγRs.

comparable levels of IgG1 and IgG4 anti-Dsg1 antibodies in the skin tissues of hFCGR-Tg mice (*Figure 4—figure supplement 3E*), suggesting both IgG1 and IgG4 anti-Dsg1 antibodies can reach skin tissues. Interestingly, when the anti-Dsg1 autoantibody dosage is reduced, it appears that PF24-9(IgG4) is not less but tends to be more pathogenic than PF24-9(IgG1) antibodies based on the histopathological symptoms (*Figure 4C*), in contrast with our analysis of anti-ADAMTS13 autoantibodies.

## Anti-Dsg1 autoantibodies are more pathogenic in the absence of FcγRs

To investigate whether Fc-FcγR interaction impacts the pathogenicity of anti-Dsg1 antibodies, FcγRα-null mice were used. Strikingly, PF24-9(IgG1) autoantibodies induced more severe ear lesions in FcγRα-null mice than in hFCGR-Tg mice (*Figure 4C*). Increased infiltration of myeloid effector cells (*Figure 4—figure supplement 4* and *Figure 4—figure supplement 5A to C*), especially inflammation-relevant neutrophils that are very sensitive to lesions and considered the first leukocyte subset arriving at lesion sites (*de Oliveira et al., 2016*; *Radoš, 2011*), was observed in the ears of FcγRα-null mice treated with either PF24-9(IgG4) or PF24-9(IgG1) autoantibodies (*Figure 4D to F*). While these results suggest that anti-Dsg1 autoantibodies induced acute inflammation together with skin lesions, they also suggest an FcγR-mediated mechanism in attenuating anti-Dsg1 autoantibodies-induced skin lesions.

## Anti-Dsg1 autoantibodies with low affinity to FcγRs are more pathogenic

To further investigate whether Fc-FcγR interactions attenuate anti-Dsg1 autoantibody pathogenicity, the N297A and GASDALIE variants (with reduced FcγR binding and enhanced activating FcγR binding, respectively [*Figure 1—figure supplement 1B, C*; *Bournazos et al., 2019*; *Sazinsky et al., 2008*]) of IgG1 anti-Dsg1 autoantibodies were produced (*Figure 4—figure supplement 2A*) and evaluated in the adult pemphigus model. Strikingly, the N297A variants of both pathogenic anti-Dsg1 clones (PF24-9 and PF1-8-15) are much more potent than their matched GASDALIE variants in inducing ear lesions (*Figure 5A*, and *Figure 5—figure supplement 1A, B*) and inflammation in hFCGR-Tg mice, as shown by increased ear thickness and weight (*Figure 5B, C*, and *Figure 5—figure supplement 1C, D*), as well as increased infiltration neutrophils (*Figure 5D, E*, *Figure 5—figure supplement 1E-G*). Histopathological analysis of skin lesions at an early time point (day 3) (*Figure 5—figure supplement 1A*) and a later time point (day 6) (*Figure 5—figure supplement 1B*) revealed that while both the PF1-8-15(N297A) and PF1-8-15(GASDALIE) variants could induce skin lesions, the PF1-8-15(GASDALIE) variant-treated mice recovered faster. Consistent results were also observed in WT mice treated with PF1-8-15(N297A) and PF1-8-15(GASDALIE) antibodies when the onset and recovery of skin lesions were monitored (*Figure 5—figure supplement 2A*). The process lasted 2–3weeks, and the recovery was confirmed by the histological and vision examinations (*Figure 5—figure supplement 2A, B*), and by quantifying the infiltrating neutrophils (*Figure 5—figure supplement 2C, D*). At the same time, the levels of serum human IgG dropped to undetectable on day 6 (*Figure 5—figure supplement 2E*), and can't be detected on day 23 in the skin tissue by DIF (*Figure 5—figure supplement 2F*). These results further support that the first few days seem the best time window for studying anti-Dsg1 autoantibody pathogenicity.

The pathogenic function of N297A and GASDALIE anti-Dsg1 autoantibodies was also evaluated in nude mice, where skin lesions can be better observed. As shown in *Figure 5F* and *Figure 5—figure supplement 1I*, PF1-8-15(N297A) autoantibodies induced more severe skin lesions and slower recovery as compared to matched PF1-8-15(GASDALIE) autoantibodies. These results further support the notion that Fc-FcγR interaction attenuates the pathogenicity of anti-Dsg1 autoantibodies, and it is worth noting that this is observed in both hFCGR-Tg and nude mice. Consistently, the

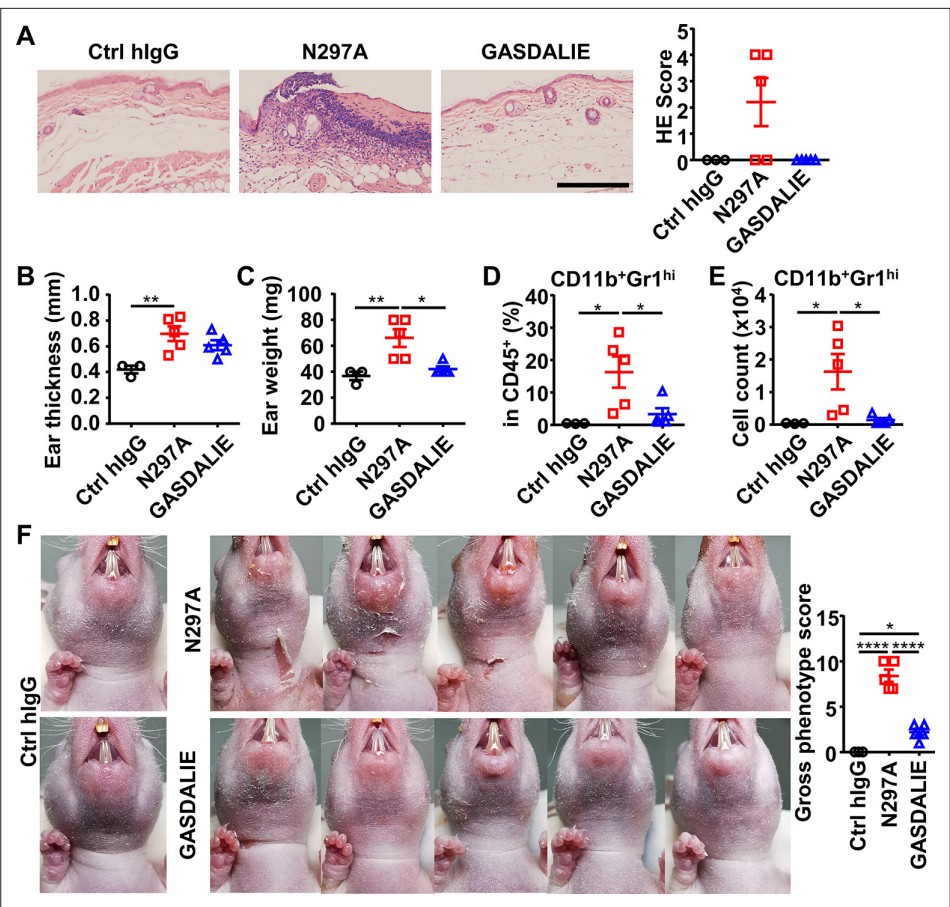

**Figure 5.** Anti-Dsg1 autoantibodies with reduced FcγR-binding are more pathogenic. (**A**) Representative photos and scores showing the hematoxylin-eosin (HE) staining results of ears of hFCGR-Tg mice 3 days after being treated with 0.5 mg of Ctrl hIgG (n=3), or PF24-9(N297A) or PF24-9(GASDALIE) autoantibodies (n=5). Scale bars: 200 μm. (**B, C**) Ear thickness (**B**) and weight (**C**) of hFCGR-Tg mice in (**A**) when sacrificed at day 3. (**D, E**) Plots showing the percentage (**D**) and cell number (**E**) of infiltrating neutrophils (CD11b+Gr1hi) among leukocytes (CD45+) in the ears of mice in (**A**). (**F**) Photos and gross phenotype scores of nude mice 2 days after being treated with 0.4 mg of Ctrl hIgG or PF1-8-15(N297A) or PF1-8-15(GASDALIE) antibodies (n=5). Each symbol is derived from an individual mouse. Mean ± SEM values are plotted. One-way ANOVA with Tukey's multiple comparisons test (**A–F**). *p<0.05, **p<0.01, ****p<0.0001.

The online version of this article includes the following source data and figure supplement(s) for figure 5:

**Source data 1.** Related to *Figure 5A*.

**Source data 2.** Related to *Figure 5B*.

**Source data 3.** Related to *Figure 5C*.

**Source data 4.** Related to *Figure 5D*.

**Source data 5.** Related to *Figure 5E*.

**Source data 6.** Related to *Figure 5F*.

**Figure supplement 1.** Anti-Dsg1 autoantibodies with low affinity to FcγRs are more pathogenic.

**Figure supplement 2.** The onset and recovery of anti-Dsg1 antibody-induced skin lesions in WT mice.

**Figure supplement 3.** Fc-FcγR interaction is protective for anti-Dsg1 autoantibodies from pathogenicity.

PF1-8-15(GASDALIE) autoantibody also induced more severe skin lesions and inflammation, as well as increased infiltrating neutrophils in FcγRα-null mice as compared in hFCGR-Tg mice (***Figure 5—figure supplement 3A-E***).

## FcγR-mediated effector function promotes the clearance of immune complexes and dead keratinocytes induced by anti-Dsg1 autoantibodies

Since both FcγR-null (N297A) and FcγR-enhanced (GASDALIE) anti-Dsg1 autoantibodies can trigger skin lesions, we reasoned that the observed exacerbation of skin lesions associated with reduced Fc-FcγR binding is due to the impact of FcγR-mediated effector function at the tissue repair stage (***Gaipl et al., 2006***; ***Nagata, 2018***). Analysis of the levels of remaining anti-Dsg1 autoantibodies in the serum of both hFCGR-Tg and nude mice showed that in the presence of FcγRs, GASDALIE anti-Dsg1 autoantibodies are depleted faster than N297A anti-Dsg1 autoantibodies (***Figure 6A and B***, ***Figure 5—figure supplement 1H*** and ***Figure 6—figure supplement 1A***), and this difference was limited in FcγRα-null mice (***Figure 6C***), suggesting more FcγR-dependent uptake of Dsg1-autoantibodies immune complexes. Consistently, the levels of Dsg1-autoantigen-autoantibody immune complexes were clearly higher in PF1-8-15(N297A)-treated nude mice than in PF1-8-15(GASDALIE)-treated nude mice, but not in FcγRα-null mice (***Figure 6D***, and ***Figure 6—figure supplement 1B***). At the same time, we detected comparable levels of N297A and GASDALIE anti-Dsg1 antibodies in the skin tissue of nude mice 6 and 24 hr after treatment (***Figure 6—figure supplement 1C***), suggesting the different pathogenic functions of these two antibodies are not due to different accessibility to the skin tissues. Notably, we also observed more epidermal blistering and dead keratinocytes at skin lesions in PF1-8-15(N297A)-treated nude mice than in PF1-8-15(GASDALIE)-treated nude mice (***Figure 6E, F*** and ***Figure 6—figure supplement 1D***). More epidermal blistering and dead keratinocytes were also observed in FcγRα-null mice regardless of whether PF1-8-15(N297A) or PF1-8-15(GASDALIE) was used (***Figure 6E, F***). These results suggest that FcγR-mediated effector function promotes the clearance of dead keratinocytes.

## FcγR-enhanced non-pathogenic anti-Dsg1 autoantibodies attenuate skin lesions induced by pathogenic anti-Dsg1 antibodies

We further hypothesized that non-pathogenic anti-Dsg1 antibodies could also promote the FcγR-mediated clearance of autoantigen-autoantibody immune complexes and, therefore, the healing of skin lesions. To test this hypothesis, a non-pathogenic but cross-reactive anti-Dsg1 autoantibody clone (PF1-2-22), previously isolated from PF patients (***Yamagami et al., 2009***), was produced as the GASDALIE variant and confirmed to have different binding epitope as pathogenic anti-Dsg1 clones PF1-8-15 and PF24-9 (***Figure 4—figure supplement 2D***, ***Supplementary file 1e***). Notably, the non-pathogenic PF1-2-22(GASDALIE) could attenuate the skin lesions induced by pathogenic PF1-8-15(N297A) autoantibodies in nude mice, as shown by the reduced skin lesions (***Figure 7A, B***) and epidermal blistering (***Figure 7C***). Furthermore, fewer apoptotic keratinocytes in the epidermis were observed in skin tissues isolated from non-pathogenic anti-Dsg1 autoantibody-treated mice (***Figure 7D***). These results suggest that the FcγR-mediated effector function can attenuate pathogenic anti-Dsg1 autoantibody-induced skin lesions by promoting the clearance of apoptotic keratinocytes that may otherwise undergo secondary necrosis and induce inflammation, and therefore contribute to the healing of Dsg1 autoantibody-induced skin lesions in PF models.

## Discussion

It is striking that the IgG4 subclass and Fc-FcγR interaction have the opposite impact on the pathogenicity of autoantibodies isolated from different IgG4-mediated autoimmune diseases. Although the IgG4 autoantibodies are pathogenic in both the TTP and PF patients and animal models, the IgG4 subclass can attenuate the pathogenic function of anti-ADAMTS13 autoantibodies when the otherwise more pathogenic IgG1 subclass is considered. This is due, at least in part, to its weak FcγR-mediated effector function since either reducing FcγR-binding affinity or ablating FcγRs can also attenuate the pathogenicity of anti-ADAMTS13 autoantibodies. In contrast, the IgG4 subclass and Fc-FcγR interaction have the opposite impact on the pathogenic function of anti-Dsg1 autoantibodies, since either

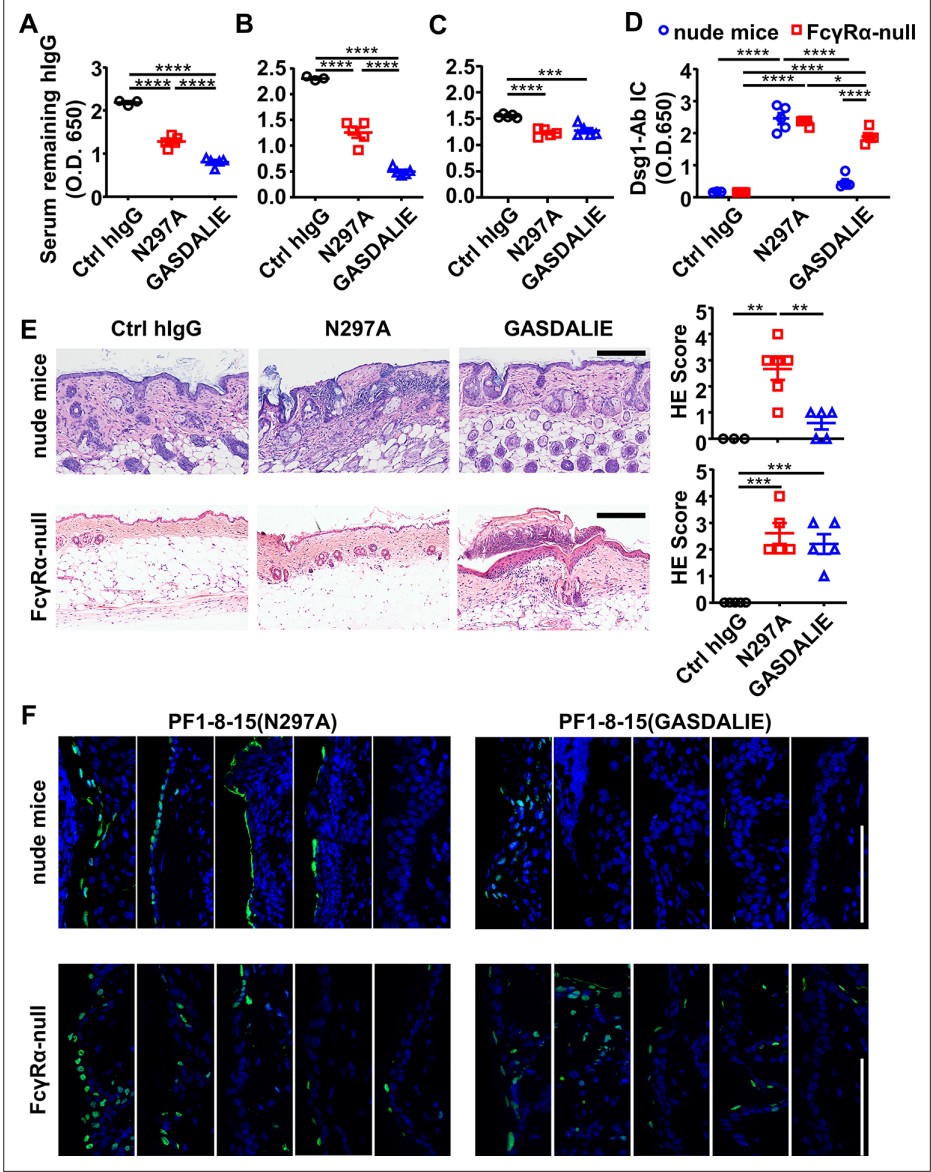

**Figure 6.** FcγR-mediated effector function promotes the clearance of immune complexes and dead keratinocytes induced by anti-Dsg1 autoantibodies. (**A–D**) Plots showing the levels of serum remaining free hIgG (**A, B, C**) and Dsg1-Ab immune complex (**D**) in mice (n=5) as treated in *Figure 5A* (**A**), or in nude mice (**B, D**) and FcγRα-null mice (**C, D**) 2 days after being treated with 0.4 or 0.5 mg of anti-Dsg1 IgG1 variants PF1-8-15(N297A) or PF1-8-15(GASDALIE), respectively. (**E**) Photos and scores showing the hematoxylin-eosin (HE) staining results of the skin of nude mice and FcγRα-null mice as treated in (**B and C**) (n=5). Scale bars: 200 μm. (**F**) Photos showing the TUNEL staining results of skin tissues collected from mice (n=5) in (**B and C**), with positive cells (green) corresponding to the epidermis. Scale bars: 100 μm. Each photo or symbol is derived from an individual mouse. Mean ± SEM values are plotted. One-way ANOVA with Tukey's multiple comparisons (**A, B, C, E**) and two-way ANOVA (**D**) with Tukey's multiple comparisons. *p<0.05, **p<0.01, ***p<0.001, ****p<0.0001.

The online version of this article includes the following source data and figure supplement(s) for figure 6:

**Source data 1.** Related to *Figure 6A*.

**Source data 2.** Related to *Figure 6B*.

**Source data 3.** Related to *Figure 6C*.

**Source data 4.** Related to *Figure 6D*.

**Source data 5.** Related to *Figure 6E*.

**Source data 6.** Related to *Figure 6E*.

*Figure 6 continued on next page*

*Figure 6 continued*

**Figure supplement 1.** FcγR-mediated effector function promotes the clearance of immune complexes and dead keratinocytes induced by anti-Dsg1 autoantibodies.

reducing FcγR-binding affinity or ablating FcγRs can exacerbate their pathogenic function. Our results provide direct evidence for the differential impact of IgG subclasses on the pathogenic function of autoantibodies.

Previously, mixed results have been reported regarding the pathogenic potential of IgG1 and IgG4 autoantibodies in both TTP and PF. On the one hand, IgG4 autoantibody-dominant TTP serum

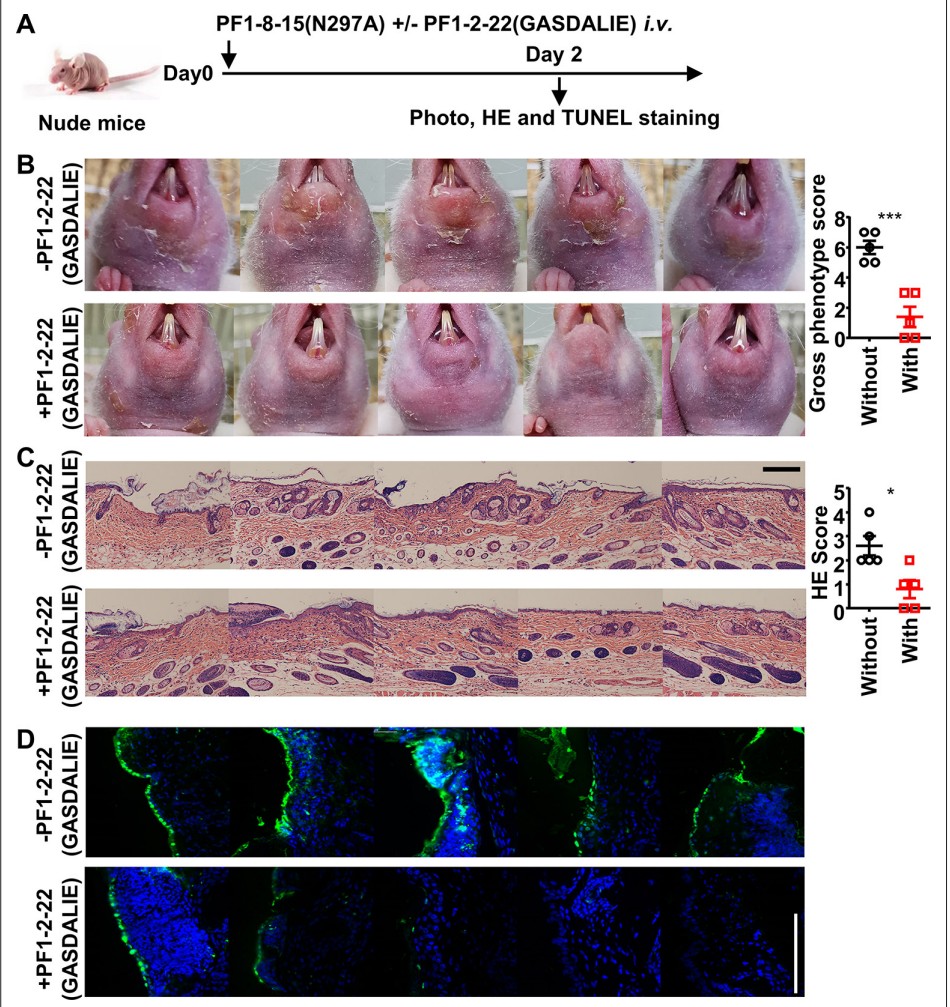

**Figure 7.** FcγR-enhanced non-pathogenic anti-Dsg1 autoantibodies attenuate skin lesions induced by pathogenic anti-Dsg1 antibodies. (**A**) Schematic diagram of the experimental design for evaluating the impact of non-pathogenic anti-Dsg1 IgG1 with GASDALIE mutations (PF1-2-22(GASDALIE)) on the pathogenicity of pathogenic anti-Dsg1 IgG1 with N297A mutation (PF1-8-15(N297A)) in nude mice. (**B–D**) Photos and gross phenotype scores showing skin lesions around the mouth (**B**), Hematoxylin-eosin (HE) staining results and scores of skin tissues (**C**), TUNEL staining results of skin (**D**) in nude mice (n=5) 2 days after being treated with 0.5 (**B**) or 0.4 mg (**C, D**) of pathogenic PF1-8-15(N297A) together with (+) or without (-) an equal amount of non-pathogenic (PF1-2-22(GASDALIE)) antibodies, with TUNEL positive cells (green) correspond to the epidermis in (D). Scale bars: 200 μm (**C**) or 100 μm (**D**). Each photo is derived from an individual mouse. Unpaired t test (**B, C**). *p<0.05, ***p<0.001.

The online version of this article includes the following source data for figure 7:

**Source data 1.** Related to *Figure 7B*.

**Source data 2.** Related to *Figure 7C*.

samples were recently reported to have a stronger inhibitory effect on ADAMTS13 activity than IgG1 autoantibody-dominant TTP serum samples (*Sinkovits et al., 2018*), despite that this difference does not seem to correlate with disease course in the same study (*Sinkovits et al., 2018*). On the other hand, patients with high levels of IgG1 and low levels of IgG4 anti-ADAMTS13 autoantibodies were reported to have high mortality rate (*Ferrari et al., 2009*); and IgG1 and IgG3, not IgG4 anti-ADAMTS13 autoantibody titers, were among the most strongly associated factors with clinical severity of acute TTP (*Bettoni et al., 2012*). Furthermore, a subclass switching from IgG1 to IgG4, but not IgG4 to IgG1 in anti-ADAMTS13 autoantibodies, was observed at the first episode/remission transition in TTP (*Sinkovits et al., 2018*). While these studies established a correlation between IgG subclass and disease severity, a direct comparison between IgG4 and IgG1 autoantibodies has not been allowed given the complexity of polyclonal anti-ADAMTS13 autoantibodies in serological studies. The majority of these studies supported our conclusion that IgG4 is less pathogenic as compared to matched IgG1 anti-ADAMTS13 autoantibodies. Also, consistently, the inverse correlation between IgG4 and IgG1 anti-ADAMTS13 autoantibodies and their association with different levels of ADAMTS13 antigen levels have been described previously in an independent study (*Ferrari et al., 2009*).

Regarding the impact of IgG subclass on autoantibody pathogenicity in PF, it has been clear that the emergence of IgG4 anti-Dsg1 autoantibodies has a stronger correlation than IgG1 autoantibodies with endemic PF disease activity (*Warren et al., 2003*). Previously, Fab fragments of anti-Dsg1 antibodies were reported to be more pathogenic as compared to intact IgG4 autoantibodies (*Rock et al., 1990*). While it suggests that the IgG Fc is not essential for anti-Dsg1 autoantibody pathogenicity, it also supports a role of IgG4 Fc in modulating the pathogenicity of anti-Dsg1 autoantibodies. Interestingly, 'non-pathogenic IgG1 anti-Dsg1 antibodies' described in endemic PF patients in the preclinical stage and HC living in the endemic areas were proposed to have non-pathogenic binding epitopes (*Aoki et al., 2015*; *Li et al., 2003*), and no functional roles, as far as we know, have been proposed for the IgG1 subclass and these 'non-pathogenic antibodies'. Our study provides direct evidence supporting a role of the IgG subclass and Fc-FcγR interaction in the pathogenic function of anti-Dsg1 autoantibodies isolated from PF patients, as well as a potentially protective role of non-pathogenic IgG1 anti-Dsg1 antibodies described previously.

The opposite impact of the IgG4 subclass and Fc-FcγR interaction on the pathogenic function of autoantibodies in anti-ADAMTS13 and anti-Dsg1 autoantibodies suggests that these autoantibodies have different modes of action in IgG4-mediated autoimmune diseases. Previously, many antibodies have been studied for their mode of action and the impact of Fc-FcγR engagement. Interestingly, different types of antibodies categorized by mode of action are impacted very differently by Fc-FcγR engagement. It has been well established that effector antibodies, which eliminate their targets (bacteria, virus, toxin, cancer cells, etc.), all require activating FcγRs to mediate their optimal activities (reviewed in *Bournazos and Ravetch, 2017*). Agonistic antibodies targeting a number of TNF receptor superfamily members (CD40, DR5, CD137, etc.) have been shown to depend on inhibitory FcγRIIB for their optimal activities (reviewed in *Beers et al., 2016*). In contrast, anti-PD1 antibodies that function by blocking PD1/PD-L1 interaction have been shown to function best when they do not bind to FcγRs (*Dahan et al., 2015*). While these findings are important for the design and engineering of antibody applications according to their modes of action, it has not been well studied whether they apply to autoantibodies, with many of which have unclear modes of action.

Our study of anti-ADAMTS13 autoantibodies suggests that they have multiple functional mechanisms. On the one hand, their enzyme inhibition ability in the absence of FcγRs suggests that anti-ADAMTS13 antibodies can function by blocking, which is consistent with their requirement of specific binding epitopes — autoantibodies targeting the spacer domain and metalloprotease domain of ADAMTS13, which are required for binding to and cleaving its substrate vWF, have been reported to be pathogenic (*Feys et al., 2010*; *Ostertag et al., 2016a*; *Ostertag et al., 2016b*; *Zheng, 2015*). On the other hand, the protective effect of the IgG4 subclass (vs. IgG1) and FcγR deficiency in our TTP model suggest that anti-ADAMTS13 autoantibodies also function at least in part as effector antibodies. Since ADAMTS13 proteins are circulating in the blood, a plausible mechanism would be that FcγR-mediated depletion of ADAMTS13 can further exacerbate the reduction of ADAMTS13 enzymatic activities. This notion is supported by the observation that IgG4-ADAMTS13 immune complexes persist longer than IgG1-ADAMTS13 immune complexes, likely due to reduced clearance (*Ferrari et al., 2014*; *Lux et al., 2013*). In this regard, it is noted that among all the natural human IgG

subclasses, IgG4 has the least effector function, suggesting that switching to the IgG4 subclass is a protective mechanism in the context of TTP.

The increased pathogenic function of anti-Dsg1 autoantibodies observed in FcγR-deficient mice (vs. FcγR-sufficient mice) and in the form of IgG variants with weak binding affinity to FcγRs (vs. IgG variants with enhanced binding affinity to FcγRs) suggest that anti-Dsg1 autoantibodies also primarily function as blocking antibodies (*Koneczny, 2018*). This is also supported by the previous finding that these antibodies require a specific binding epitope, and they can function in the form of scFv (*Ishii et al., 2008*; *Yamagami et al., 2009*; *Yoshida et al., 2017*). However, it is intriguing that Fc-FcγR interaction attenuates anti-Dsg1 autoantibody-induced pathogenicity, which suggests that either FcγR-dependent effector function or agonistic function has a contribution to the overall effect of anti-Dsg1 autoantibodies. The agonistic function is not likely given that Dsg1-signaling has been suggested to be a pathogenic mechanism (*Hammers and Stanley, 2020*; *Spindler et al., 2018*). Since anti-Dsg1 autoantibody-induced skin lesions belong to the type II hypersensitivity, in which the keratinocytes are attacked, and inefficient clearance of apoptotic cells may trigger secondary necrosis and delay the healing, our results support that FcγR-mediated effector function is beneficial for tissue repair and wound healing by contributing to the clearance of Dsg1-autoantibody immune complexes. It will be interesting to investigate whether other autoantibodies are also involved in the clearance of autoantigen and cell debris containing autoantigens.

While our study of anti-ADAMTS13 autoantibodies supports the notion that switching to immune-inert IgG4 subclass is a protective mechanism in IgG4-mediated autoimmune diseases, our analysis of anti-Dsg1 autoantibodies suggests the opposite. It appears that the distinct impact of IgG subclass and Fc-FcγR engagement on different antibodies with blocking function lies in the effect of their effector functions in their respective context. In the previously described blocking anti-PD1 antibodies (*Dahan et al., 2015*), either FcγR-dependent depletion of PD1-expressing T cells or agonistic function of anti-PD1 antibodies results in the reduction of T cell immunity, which counters the effect of blocking PD1 signal in T cells. In the case of anti-ADAMTS13 autoantibodies, FcγR-mediated depletion of ADAMTS13 synergizes with the blocking function of anti-ADAMTS13 autoantibodies to further reduce the enzymatic activity of ADAMTS13. In the case of anti-Dsg1 autoantibodies, the FcγR-mediated effector function may indirectly counter the pathogenic function of anti-Dsg1 autoantibodies by promoting the clearance of apoptotic cells containing autoantigens and tissue repair. Therefore, carefully examining the impact of IgG subclasses and Fc-FcγR interactions on autoantibody functions is critical for understanding their mode of action in the context of different biological processes and disease settings.

# Materials and methods

**Key resources table**

| Reagent type (species) or resource | Designation | Source or reference | Identifiers | Additional information |
|---|---|---|---|---|
| Cell line (*Homo sapiens*) | HEK293S | Expression Systems, courtesy of Dr Ai-Wu Zhou (Shanghai Jiao Tong University School of Medicine) | RRID:CVCL_A784 | |
| Cell line (*Spodoptera frugiperda*) | Sf9 | Expression Systems, courtesy of Dr Ai-Wu Zhou (Shanghai Jiao Tong University School of Medicine) | RRID:CVCL_0549 | |
| Antibody | Anti-mouse CD45.2 Alexa Fluor 700 (Mouse monoclonal) | BD | Cat# 560693 RRID:AB_1727491 | FACS (1:200) |
| Antibody | Anti-mouse CD11b APC (Rat monoclonal) | eBioscience | Cat# 17-0112-82 RRID:AB_469343 | FACS (1:500) |
| Antibody | Recombinant Anti-Human IgG antibody (Rabbit monoclona) | Abcam | Cat# ab109489, RRID:AB_10863040 | IF (1:150) |
| Antibody | Anti-Rabbit IgG (H+L) Antibody, Alexa Fluor 488 (Donkey polyclonal secondary) | Thermo Fisher Scientific | Cat# A-21206, RRID:AB_2535792 | IF (1:400) |

*Continued on next page*

*Continued*

| Reagent type (species) or resource | Designation | Source or reference | Identifiers | Additional information |
|---|---|---|---|---|
| Antibody | Anti-Lambda Light Chain Monoclonal Antibody, Biotin (Mouse monoclonal) | BD Biosciences | Cat# 555794, RRID:AB_396129 | ELISA (1:2000) |
| Antibody | Anti-Dsg1 TTP1-8-15(IgG1) Biotin (human monoclonal) | This paper | | ELISA (1 µg/ml) |
| Antibody | Anti-Human IgG-Fc Fragment Antibody HRP (Goat polyclonal) | Bethyl | Cat# A80-104P, RRID:AB_67064 | ELISA (1:10,000) |
| Antibody | Anti-Human IgG1 Hinge-HRP (Mouse monoclonal) | SouthernBiotech | Cat# 9052–05, RRID:AB_2796619 | ELISA (1:3160) |
| Antibody | Anti-Human IgG4 Fc-HRP (Mouse monoclonal) | SouthernBiotech | Cat# 9200–05, RRID:AB_2796691 | ELISA (1:3160) |
| Antibody | Anti-Human IgG1 Fc Secondary Antibody, HRP (Mouse monoclonal) | Thermo Fisher Scientific | Cat# MH1715, RRID:AB_10374315 | ELISA (1:2000) |
| Antibody | Anti-Human IgG4 Fc Secondary Antibody, HRP (Mouse monoclonal) | Thermo Fisher Scientific | Cat# MH1742, RRID:AB_2539714 | ELISA (1:2000) |
| Antibody | Anti-Human IgG (H+L) (Goat polyclonal) | Jackson ImmunoResearch Labs | Cat# 109-005-088, RRID:AB_2337539 | ELISA (2 µg/ml) |
| Antibody | Anti-Human IgG-heavy and light chain Antibody HRP (Goat polyclonal) | Bethyl | Cat# A80-219P, RRID:AB_67076 | ELISA (1:10,000) |
| Peptide, recombinant protein | Streptavidin-HRP | BD Biosciences | Cat# 554066, RRID:AB_2868972 | ELISA (1:1000) |
| Peptide, recombinant protein | Recombinant Human ADAMTS13 (Full Length) Protein, CF | R&D Systems | Cat# 6156-AD | ELISA (2 µg/ml) |
| Commercial assay or kit | Human ADAMTS13 Quantikine ELISA Kitg | R&D Systems | Cat# DADT130 | |
| Chemical compound, drug | FRETS-VWF73 | Anaspec | Cat# 63728–01 | |
| Chemical compound, drug | Dispase II | Roche | Cat# 4942078001 | |
| Chemical compound, drug | DNase I | Sigma | Cat# D5025 | |
| Chemical compound, drug | Collagenase IV | Sigma/biosharp | Cat# C5138 | |
| Software, algorithm | GraphPad Prism | GraphPad Prism | RRID:SCR_002798 | |
| Other | DAPI stain | Invitrogen | DD3571 | (0.5 µg/ml) |

## Patients and samples collection

We analyzed plasma samples of 44 acquired TTP patients investigated in Jiangsu Institute of Hematology, The First Affiliated Hospital of Soochow University, Jiangsu, China, between September 2019 and April 2020. This institute is providing diagnostic services for patients in China suspected of having thrombotic microangiopathies. The diagnosis of acquired TTP was based on the following criteria: (1) thrombocytopenia (platelet count below 150 g/l) and hemolytic anemia (Coombs-negative anemia, elevated LDH); (2) deficient ADAMTS13 activity (<5%, measured by R-CBA assay [the residual collagen-binding activity], as described below); and (3) detectable inhibitory anti-ADAMTS13 autoantibodies as analyzed by the R-CBA method (*Gerritsen et al., 1999*). Blood samples were collected during an acute episode and anticoagulated with sodium citrate before plasma exchange therapy. Plasma samples were separated by centrifugation and stored at −70°C until measurements. Sodium citrate-anticoagulated plasma samples were used for the determinations of ADAMTS13 activity, ADAMTS13 Ag levels, and autoantibody subclasses and concentrations.

To study the impact of IgG subclass on the pathogenicity of anti-Dsg1 autoantibodies, 53 PF patients, including 21 patients at stable stage (defined as no new skin lesions and erosions for at least

1 month, and with gradually reducing corticosteroid dosage) and 32 patients at active stage (including newly diagnosed patients without any treatment or stable patients developing new lesions, lasting more than 1 week), from Department of Dermatology, Rui Jin Hospital, Shanghai Jiao Tong University School of Medicine, China, were recruited and analyzed. All PF patients exclusively had anti-Dsg1 autoantibodies, but not anti-Dsg3 autoantibodies, as detected by ELISA. Serum and sodium citrate-anticoagulated plasma of healthy people was obtained from the medical center or healthy blood donors.

## Mice

Adult and neonatal C57BL/6 WT mice and Balb/c nude mice were obtained from SLAC (Shanghai, China). FcγR-deficient mouse ($Fcgr1^{-/-}Fcgr2b^{-/-}Fcgr3^{-/-}Fcgr4^{-/-}$, referred to as 'FcγRα-null') (*Smith et al., 2012*), FcγR-humanized mouse ($Fcgr1^{-/-}Fcgr2b^{-/-}Fcgr3^{-/-}Fcgr4^{-/-}/FCGR1A^+/FCGR2A^{R131+}/FCGR2B^{I232+}/FCGR3A^{F158+}/FCGR3B^+$, referred to as 'hFCGR-Tg') (*Smith et al., 2012*), and Fc receptor common γ-chain-deficient mouse ($Fcer1g^{-/-}$) (*Clynes et al., 1998b*; *Takai et al., 1994*) have been described elsewhere and were kindly provided by Dr Jeffrey Ravetch (The Rockefeller University). hFCGR-Tg or FcγRα-null mice produced by breeding or by bone marrow reconstitution were used. The method to generate bone marrow chimeric mice has been described previously (*Liu et al., 2019*). Briefly, 8–10 weeks female C57BL/6 WT mice (SLAC, Shanghai, China) were lethally irradiated with 8 Gy X-ray using RS 2000pro X-ray biological Irradiator (Rad Source Technologies, Inc, Buford, GA), and $2\times10^6$ bone marrow cells of hFCGR-Tg or FcγRα-null mice were transferred to these irradiated mice through tail vein injection. Successful bone marrow reconstitution was confirmed 2 months later by analyzing FcγRIIB expression in B cells and CD11b$^+$ myeloid cells in peripheral blood by flow cytometry. Mice were used at the age of 8–12 weeks or 2–4 months after bone marrow reconstruction used unless stated otherwise.

## Cell lines

In this study we used authentic and mycoplasma-free cells that were provided by Dr. Aiwu Zhou (Shanghai Jiao Tong University School of Medecine). Sf9 cells were cultured at 27°C in serum-free SIM SF medium (Sino Biological Inc). HEK293S cells were cultured at 37°C in 5% $CO_2$. No additional authentication was performed by the authors of this study. Cell line was negative for mycoplasma. No commonly misidentified lines were used in this study. All cell lines were kept at low passages in order to maintain their health and identity.

## Antibodies

The amino acid sequences of the variable region of TTP1-420 anti-ADAMTS13 autoantibody were obtained from a published paper (*Ostertag et al., 2016b*). Nucleic acid sequences of the variable region of pathogenic PF1-8-15 and PF24-9 anti-Dsg1 scFv and non-pathogenic PF1-2-22 anti-Dsg1 scFv were obtained from the patent (Patent No.: US8846867B2). Antibody heavy chain variable sequences and light chain sequences were synthesized and inserted into the pFL_DEC expression vector with or without human IgG constant region (IgG1 or IgG4), respectively, as described previously (*Liu et al., 2019*). IgG1 Fc variants with specific mutations (N297A or G236A/S239D/A330L/I332E) were generated by site-directed mutagenesis. Paired antibody heavy and light chain expression vectors were used to transiently co-transfect HEK293S cells. IgG antibodies in the supernatant were collected several days later and purified with protein G Sepharose (GE Healthcare), and dialyzed to PBS and stored at 4°C. LPS (endotoxin) levels were analyzed by the Limulus amebocyte lysate assay (Thermo Fisher Scientific) and confirmed to be <0.01 EU/μg.

## Production of hDsg1 and hDsg2 extracellular domains and their chimeric proteins

DNA fragments encoding the signal peptide, pro-sequence, and entire extracellular domains of Dsg1 (GenBank accession no. NM_001942) and Dsg2 (GenBank accession no. NM_001943) were obtained by reverse transcription polymerase chain reaction (RT-PCR, Thermo Fisher Scientific) amplification using RNA extracted from healthy human skin (obtained from plastic surgery) by TRIzol reagent (Invitrogen) as the template. DNA fragments encoding of Dsg1/Dsg2 chimeric proteins consisting of various extracellular domain segments of Dsg1 were obtained by overlapping PCR using Dsg1 and

Dsg2 vectors as the templates. The forward and reverse primers used are listed in **Supplementary file 1f**. BamH1 and Sal1 are added to 5' and 3' end primers, respectively, and were used for subcloning the DNA fragments encoding of Dsg1/Dsg2 chimeric proteins into the pFastBac1 vector that was engineered to contain the FlagHis tag (dykddddkfvehhhhhhhh) sequence between the Sal1 and Not1 sites. The recombinant donor plasmid was used to transform competent DH10Bac *Escherichia coli* cells, after which blue-white plaque assay was performed to confirm successful site-specific transposition. Dsg1, Dsg2, and Dsg1/Dsg2 chimeric proteins were expressed in the Sf9 cells and purified as previously described (**Ding et al., 1999**).

## ELISA

To measure the binding ability of human IgG and Fc variants to mouse FcγRs, a previously described protocol (**Liu et al., 2019**) was modified. Briefly, 100 µl of 2 µg/ml TTP1-420 anti-ADAMTS13 antibodies with various constant domains (IgG1/IgG4/N297A/GASDALIE) were coated in 96-well MaxiSorp flat plate (Nunc) at 4°C overnight. After discarding the liquid and washing with PBS containing 0.05% Tween-20 (PBST), the plates were blocked with 200 µl of 1% or 2% BSA at room temperature (RT) for 2 hr and washed two times with PBST, after which 100 µl of serially diluted (0.001–1 µg/ml, 1:3.16) biotinylated mouse FcγRs (Sino Biological) were added and incubated at RT for 1 hr. Plates were then washed three times with PBST and further incubated with 100 µl of diluted Streptavidin-HRP (1:1000, BD Pharmingen) for 1 hr, and followed by washing four times and developing with 100 µl of TMB peroxidase substrate (KPL). The absorbance at 650 nm was recorded using a multifunctional microplate reader (SpectraMaxi3, Molecular Devices).

Similar protocols were applied to other ELISA analyses with except that when Dsg1, Dsg2, or Dsg1/2 chimeric proteins were involved, all reagents were dissolved or diluted in TBS-Ca buffer (TBS buffer with 1 mM CaCl$_2$) and TBS-Ca-T (TBS-Ca containing 0.05% Tween-20) was used as washing buffer, as well as the following:

1. To detect the binding kinetics of TTP1-420 anti-ADAMTS13 or PF1-8-15 anti-Dsg1 antibodies with different constant domains (IgG1/IgG4/N297A/GASDALIE) to their antigens, respectively, 100 µl of 2 µg/ml ADAMTS13 (R&D Systems, diluted with bicarbonate/carbonate buffer containing 15 mM Na$_2$CO$_3$ and 35 mM NaHCO$_3$, pH = 9.6) and 100 µl of 5 µg/ml recombinant Dsg1 was coated. Serially diluted specific and Ctrl hIgG (Jackson ImmunoResearch Laboratories) (0.00316–3.16 µg/ml) were analyzed; TTP1-420(IgG1) was used as irrelevant IgG1 control for anti-Dsg1 antibodies. Biotinylated mouse anti-human lambda chain (1:2000, Clone JDC-12 [RUO], BD Pharmingen) was used as detecting antibody.
2. To compare the binding epitopes of different anti-Dsg1 antibody clones, different anti-Dsg1 IgG1 antibody clones (PF1-8-15, PF24-9, or PF1-2-22) or TBS-Ca as negative control were coated; recombined Dsg1 (1 µg/ml) was analyzed; biotinylated PF1-8-15(IgG1) (1 µg/ml), or αCD40(IgG1) (1 µg/ml, as negative control) (EZ-Link Micro Sulfo-NHS-Biotinylation Kit [Thermo Fisher Scientific]) was used as detecting antibody.
3. To determine the total IgG, IgG1, and IgG4 anti-ADAMTS13 antibodies in TTP and HC plasma samples, 100 µl of 2 µg/ml ADAMTS13 was coated; sodium citrate-anticoagulated plasma (dilution: 1–100 for IgG, 1–10 for IgG1 and IgG4) were analyzed; serially diluted TTP1-420(IgG1) and TTP1-420(IgG4) antibodies were used as standards of IgG1 and IgG4, respectively; HRP-conjugated goat anti-human IgG Fc (1:10,000, Bethyl Laboratories), mouse anti-human IgG1 (1:3160, 4E3, SouthernBiotech) and IgG4 (1:3160, HP6025, SouthernBiotech) were used as detecting antibodies. IgG1 and IgG4 anti-ADAMTS13 antibody levels were calculated based on their respective standard curves and normalized to HC controls ([TTP anti-ADAMTS13 levels]/[HC anti-ADAMTS13 average levels]) (considering the background caused by non-specific IgG in plasma). Based on the relative normalized IgG1 and IgG4 anti-ADAMTS13 antibody levels, TTP patients were divided into 'IgG1-dominant TTP' and 'IgG4-dominant TTP' groups.
4. To determine the total IgG, IgG1, and IgG4 anti-Dsg1 antibodies in PF and HC serum samples, 100 µl of 5 µg/ml recombinant Dsg1 was coated, and diluted serum samples (dilution: 1–100 for IgG1, 1–1000 for IgG4 and IgG) were analyzed, together with serially diluted PF24-9(IgG1) and PF24-9(IgG4) antibodies as references (0.316 µg/ml as positive controls [PC]); HRP-conjugated goat anti-human IgG Fc (1:10,000, Bethyl Laboratories), mouse anti-human IgG1 (1:2000, HP6070, Thermo Fisher Scientific) and IgG4 (1:2000, HP6023, Thermo Fisher Scientific) were used as detecting antibodies; unit values of Dsg1-specific IgG, IgG1, and IgG4 antibodies were calculated according to the commercial anti-Dsg1 IgG detecting kit (Medical & Biological

Laboratories) using the following formula: $(OD650_{sample}-OD650_{HC})/(OD650_{PC}-OD650_{HC})$*Dilution factor.

5. To analyze the binding epitopes of anti-Dsg1 IgG1 and IgG4 antibodies in PF serum, recombined human Dsg1, Dsg2, and Dsg1/2 chimeric proteins (100 µl of 5 µg/ml) were coated and 1:100 diluted serum samples were analyzed; HRP-conjugated mouse anti-human IgG1 (1:2000, HP6070, Thermo Fisher Scientific) and IgG4 (1:2000, HP6023, Thermo Fisher Scientific) were used as detecting antibodies.

6. To detect the levels of free hIgG in mouse serum samples, 100 µl of 2 µg/ml goat anti-human IgG (H+L) (Jackson ImmunoResearch Laboratories) was coated and serum samples diluted to optimized concentration were analyzed (1:100 or 1:1000); goat anti-human IgG (H+L) HRP (1:10,000, Bethyl Laboratories) or biotinylated mouse anti-human lambda chain (1:2000, JDC-12 [RUO], BD Pharmingen) combined with Streptavidin-HRP (1:1000, BD Pharmingen) were used as detecting antibodies for Ctrl hIgG and anti-Dsg1 IgG1 variants.

7. To detect the levels of Dsg1-specific immune complexes (IC), non-pathogenic PF1-2-22(IgG4) (5 µg/ml) was coated at 37°C for 6 hr; diluted serum samples (1:10 or 1:100) were analyzed; mouse anti-hIgG1 HRP (1:3160, 4E3, SouthernBiotech) was used as detecting antibody.

## Fab preparation

Purified PF24-9(IgG1) antibody was digested to produce Fab fragments using immobilized papain (Pierce). Briefly, 0.125 ml of immobilized papain agarose was equilibrated with 0.5 ml of digestion buffer containing cysteine•HCl. Then, 4 mg desalinated PF24-9(IgG1) was mixed with equilibrated immobilized papain. Papain digestion was allowed to proceed for 7 hr, shaking vigorously at 37°C. The reaction was stopped by centrifugation to separate the antibody and papain agarose, and the degree of cleavage was evaluated by SDS-PAGE.

## Surface plasmon resonance

SPR experiments were performed with a Biacore T200 SPR system (Biacore, GE Healthcare) using a published protocol (*Tao et al., 2019*). In brief, experiments were performed at 25°C in PBS with 0.05% Tween-20. His-tagged soluble mouse FcγRs (Novoprotein) were immobilized on CM5 chips by amine coupling. Twofold serially diluted (7.8–2000 nM) human IgG1, IgG4, and IgG1 variants (clone TTP1-420) were injected through flow cells for 120 s at a flow rate of 30 µl/min for association followed by a 6 min dissociation phase. After each assay cycle, the sensor surface was regenerated with a 30 s injection of NaOH of optimized concentration at a flow rate of 50 µl/min. Background binding to blank immobilized flow cells was subtracted, and affinity constant $K_D$ values were calculated using the 1:1 binding kinetics model built in the Biacore T100 Evaluation Software (version 1.1).

## ADAMTS13 activity assays (FRETS-VWF73)

To analyze mouse ADAMTS13 activity, a published protocol (*Kokame et al., 2005*; *Ostertag et al., 2016b*) was used with modification. Briefly, 4.8 µl of plasma sample was diluted with 25.2 µl assay buffer (5 mM Bis-Tris, 25 mM $CaCl_2$, 0.005% Tween 20, pH = 6). Ten µl of the diluted sample was then transferred to a 384-well white plate (Cisbio), and mixed with 10 µl of diluted FRETS-VW73 substrate (4 µM, Anaspec). Related fluorescence units (RFU) of cleaved FRETS-VWF73 were measured for 1 hr using a multi-mode microplate reader (Synergy H1, BioTek) with the following setting: excitation at 340 nm and emission at 450 nm. RFU were recorded for 1 hr, and their changing rates over time were calculated and expressed as 'RFU/min'.

## ADAMTS13 activity assay (R-CBA)

ADAMTS13 activity of human plasma was assayed by evaluating CBA as previously described with modifications (*Yue et al., 2018*). In brief, 50 µl of sodium citrate-anticoagulated plasma samples from patients and HC were placed in Slide-A-Lyzer mini dialysis units (Pierce) and immersed in dialysis buffer (5 mM Tris-HCl, 0.1% Tween-20, and 1.5 M urea, pH = 8.3). Dialysis was performed at 37°C for 3 hr. An equal volume of the same sample was removed before dialysis and kept at RT during the dialysis as a control. The collagen type III binding capacities of the samples were then detected by ELISA. The data were analyzed as the fraction of CBA remaining after dialysis compared with the CBA of the individuals' baseline samples. One hundred percent minus the residual CBA was regarded as the ADAMTS13 activity.

## ADAMTS13 antigen quantification

ADAMTS13 antigen levels in the plasma of HC and TTP patients were measured using Human ADAMTS13 Quantikine ELISA Kit (R&D Systems) according to the manufacturer's instructions with minor modifications: (1) diluted plasma samples of HC (1:50) and TTP patients (1:2) were used; (2) the range of the standard curve was broadened to 0.78125–100 ng/ml.

## The activity of anti-ADAMTS13 antibodies in mice

WT C57BL/6, hFCGR-Tg, FcγRα-null, or *Fcer1g*[-/-] mice were treated with 10 µg per mouse of TTP1-420 anti-ADAMTS13 antibodies with different constant domains (IgG1/IgG4/N297A/GASDALIE) or control hIgG (Jackson ImmunoResearch Laboratories) on day 0 through tail vein injection. Blood samples were drawn at the time points described in the results and anticoagulated with 4% sodium citrate. Plasma was obtained after centrifugation and stored at –20°C for several days before analyzing. ADAMTS13 activity was analyzed using the VWF73-FRET method described above.

## Mouse model of PF

WT C57BL/6 neonatal mice born within 48 hr (SLAC, Shanghai, China) were subcutaneously injected with the same amount of control hIgG (Jackson ImmunoResearch Laboratories), IgG1 or IgG4 anti-Dsg1 autoantibodies in 50 µl (15.8 µg/mouse for PF24-9 and 10 µg/mouse for PF1-8-15) and euthanized 7 hr later to collect skin samples for hematoxylin-eosin (HE) staining.

Adult hFCGR-Tg and FcγRα-null mice were treated with pathogenic anti-Dsg1 antibodies and control hIgG through tail vein injection at the dosage described in the results on day 0. Skin or ear thickness was measured by caliper; orbital blood was collected to prepare serum. After mice were euthanized, one ear was digested for flow cytometry analysis, and the other ear or skin tissue was subjected to HE staining, direct immunofluorescence (DIF), and TUNEL staining.

To compare the pathogenicity of PF1-8-15(N297A) and PF1-8-15(GASDALIE) anti-Dsg1 antibodies, 8- to 10-week-old female nude mice (SLAC, Shanghai, China) were treated with 0.4–0.5 mg/mouse antibodies (as specified in figure legends) via tail vein injection on day 0. 6, 24, 48, or 96 hr later (as specified in figure legends), photographs were taken to record skin lesions, and blood was drawn to prepare serum. The levels of free anti-Dsg1 antibodies and Dsg1-specific ICs were analyzed in serum samples. Skin samples were harvested for HE, DIF, and TUNEL assay after mice were sacrificed. To study the impact of non-pathogenic anti-Dsg1 antibodies (PF1-2-22) optimized for Fc-FcγR interaction, pathogenic PF1-8-15(N297A) anti-Dsg1 antibodies (0.4–0.5 mg/mouse, as specified in figure legends) were injected to 7- to 9-week-old male nude mice (SLAC, Shanghai, China) with or without an equal amount of non-pathogenic PF1-2-22(GASDALIE) antibodies. Cutaneous lesions were recorded on day 2, and skin samples were harvested for HE, DIF, and TUNEL assay after mice were sacrificed.

## Direct immunofluorescence

Paraffin-embedded sections were used. After the sections were deparaffinized, 5-µm-thick sections were prepared. The microwave antigen repair technique was performed with 3% $H_2O_2$ and 0.01 M citrate buffer (pH = 6). After blocking with goat serum for 1 hr, sections were stained with rabbit anti-hIgG (1:150, Abcam) overnight at 4°C followed by Alexa Fluor 488-conjugated Goat Anti-Rabbit IgG (H+L) Antibody (1:400, Thermo Fisher Scientific) for 45 min at RT.

## HE and gross phenotype score

Histological scoring: semi-quantitative grading based on the percentage of skin lesions (blisters and inflammation) in the sample (*Mahoney et al., 1999*). The scores are as follows: 0, no lesions; 1, lesions only appear at the edges of the tissue (<25%); 2, local lesions, accounting for 25–50% of the tissue; 3, large lesions, accounting for 50–75% of the tissue; 4, very extensive lesions, >75% of the tissues.

Gross phenotype score: an easy grading system similar to PDAI (Pemphigus Disease Area Index) according to the area of skin lesions.

## Flow cytometry analysis

Mouse ears were cut and split into dorsal/ventral surfaces with forceps and digested with 2 ml of dispase II solution (2.5 mg/ml in PBS with 2% FBS, Roche) in six-well plate at 37°C for 90 min with shaking. After separating dermis from the epidermis using forceps, tissues (both epidermis and

dermis) were cut into tiny pieces and put into RPMI 1640 complete medium (RPMI 1640, 10% FBS, 1% Pen/Strep) containing 0.5 mg/ml of collagenase IV (Sigma/biosharp) and 100 U/ml of DNase I (Sigma) for incubating at 37°C for 60 min to complete the digestion. The digested tissue was then passed through a 70 µm cell strainer, and the debris was ground and washed through the cell strainer using 35 ml of cold PBS. Cells were collected and resuspended with 600 µl FACS buffer (PBS buffer with 0.5% FBS and 2 mM EDTA). Half of the resuspended ear cells were stained with 1 µg/ml of Alexa Fluor 700-conjugated mouse anti-mouse CD45.2 (104, BD), APC-conjugated rat anti-mouse CD11b (M1/70, eBioscience), and FITC-conjugated rat anti-mouse Gr1 (RB6-8C5, eBioscience). DAPI (0.5 µg/ml, Invitrogen) and CountBright Absolute Counting Bead (3 µl/sample, Life Technologies) were added to resuspend cells before analyzing using a BD LSRFortessa X-20 analyzer (BD Biosciences). Data were analyzed using FlowJo X. Gating strategy was shown in *Figure 4—figure supplement 4*.

## TUNEL assay

TUNEL staining was conducted per manufacturer's instructions (Roche). Images were obtained via OLYMPUS BX51 Confocal Microscope outfitted with a ×10 or ×40 objective. Apoptotic cells were defined as cells possessing a TUNEL positive (green) pyknotic nucleus.

## Statistics

Statistical analyses were performed with GraphPad Prism 6.0 or GraphPad Prism 8.3, and p values less than 0.05 were considered to be statistically significant. Asterisks indicate statistical difference within two interested groups on the figures (*$p < 0.05$. **$p < 0.01$, ***$p < 0.001$, ****$p < 0.0001$).

## Study approval

Ethical approval was obtained from the Ethics Committees in The Rui Jin Hospital of Shanghai Jiao Tong University School of Medicine and The First Affiliated Hospital of Soochow University, respectively. All PF and TTP patients and healthy volunteers signed informed consent. All mice were bred and maintained under specific pathogen-free conditions, and all animal experiments were performed under the institutional guidelines of the Shanghai Jiao Tong University School of Medicine Institutional Animal Care and Use Committee (Protocol Registry Number: A-2015–014).

# Acknowledgements

We thank Dr Jeffrey Ravetch of The Rockefeller University for providing FcγR-deficient mouse (FcγRα-null), FcγR-humanized mouse (hFCGR-Tg) and Fc receptor common γ-chain deficient mouse (*Fcer1g*$^{-/-}$), and pFL_DEC expression vector. We acknowledge the assistance of staff in the Department of Laboratory Animal Science, Shanghai Jiao Tong University School of Medicine and Shanghai Institute of Immunology. This work was supported by NNSFC projects No. 31870924, 31422020 and 31861143030. YMZ is supported by NNSFC project No. 81873431 and Jiangsu Provincial Natural Science Foundation No. BK20181164. YZ is supported by Shanghai Sailing Program No. 16YF1409700. HZ is supported by Shanghai Municipal Natural Science Foundation project No. 15ZR1436400 and Shanghai Young Oriental scholar program 2015 by Shanghai Municipal Education Commission. FL, YZ, and HZ are also supported by the innovative research team of high-level local universities in Shanghai (SHSMU-ZDCX-20210600, SHSMU-ZDCX-20210301), and Shanghai Collaborative Innovation Center of Cellular Homeostasis Regulation and Human Diseases. YMZ is also supported by The Jiangsu Provincial Key Medical Center (YXZXA2016002) and the Priority Academic Program Development of Jiangsu Higher Education Institutions (PAPD). MP is supported by NNSFC projects No. 81730085, 82173407. SZ is supported by NNSFC project No. 81903210. FL is also supported by Science and Technology Commission of Shanghai Municipality (Project No. 22140903000).

# Additional information

## Competing interests

Yanxia Bi, Yan Zhang, Huihui Zhang, Fubin Li: A patent application based on the study has been submitted (Chinese patent application number: 202011408005.X), and Fubin Li, Yanxia Bi, Yan Zhang, and Huihui Zhang are listed as inventors. The other authors declare that no competing interests exist.

## Funding

| Funder | Grant reference number | Author |
| --- | --- | --- |
| National Natural Science Foundation of China | 31870924, 31422020, 31861143030 | Fubin Li |
| National Natural Science Foundation of China | 81873431 | Yiming Zhao |
| Jiangsu Provincial Natural Science Foundation | BK20181164. | Yiming Zhao |
| Shanghai Sailing Program | 16YF1409700 | Yan Zhang |
| Shanghai Municipal Natural Science Foundation project | 15ZR1436400 | Huihui Zhang |
| Shanghai Young Oriental scholar program 2015 | | Huihui Zhang |
| Innovative research team of high-level local universities in Shanghai | SHSMU-ZDCX-20210600 | Fubin Li Yan Zhang Huihui Zhang |
| National Natural Science Foundation of China | 81730085, 82173407 | Meng Pan |
| National Natural Science Foundation of China | 81903210 | Shengru Zhou |
| The Jiangsu Provincial Key Medical Center | YXZXA2016002 | Yiming Zhao |
| The Priority Academic Program Development of Jiangsu Higher Education Institutions (PAPD) | | Yiming Zhao |
| Science and Technology Commission of Shanghai Municipality | 22140903000 | Fubin Li |

The funders had no role in study design, data collection and interpretation, or the decision to submit the work for publication.

## Author contributions

Yanxia Bi, Formal analysis, Validation, Investigation, Methodology, Writing – original draft, Project administration; Jian Su, Jianrong Xu, Resources, Investigation, Methodology; Shengru Zhou, Yingjie Zhao, Mingdong Liu, Methodology; Yan Zhang, Huihui Zhang, Funding acquisition, Methodology; Aiwu Zhou, Resources, Methodology; Meng Pan, Resources, Supervision, Validation, Writing – review and editing; Yiming Zhao, Resources, Supervision, Funding acquisition, Validation, Writing – review and editing; Fubin Li, Conceptualization, Resources, Supervision, Funding acquisition, Validation, Writing – original draft, Project administration, Writing – review and editing

## Author ORCIDs

Yanxia Bi https://orcid.org/0000-0001-7705-4777
Meng Pan https://orcid.org/0000-0003-4947-9369
Fubin Li https://orcid.org/0000-0001-6268-3378

## Ethics

Human subjects: Ethical approval was obtained from the Ethics Committees in The Rui Jin Hospital of Shanghai Jiao Tong University School of Medicine and The First Affiliated Hospital of Soochow University, respectively. All PF and TTP patients and healthy volunteers signed informed consent.

All mice were bred and maintained under specific pathogen-free (SPF) conditions, and all animal experiments were performed under the institutional guidelines of the Shanghai Jiao Tong University School of Medicine Institutional Animal Care and Use Committee (Protocol Registry Number: A-2015-014).

Decision letter and Author response
Decision letter https://doi.org/10.7554/eLife.76223.sa1
Author response https://doi.org/10.7554/eLife.76223.sa2

## Additional files

### Supplementary files

• Supplementary file 1. Additional information on relevant antibodies and patient samples. (a) Affinities of mouse FcγRs for human IgG4, IgG1, and its variants. (b) FcγR-binding properties of human IgG1 and its variants. (c) Demographic characteristics and laboratory findings of thrombotic thrombocytopenic purpura (TTP) patients. (d) Demographic characteristics and laboratory findings of pemphigus foliaceus (PF) patients. (e) Characteristics of anti-Dsg1 mAbs. (f) Polymerase chain reaction (PCR) primers used for cloning Dsg1, Dsg2, and Dsg1/Dsg2 chimeric molecules.

• Transparent reporting form

### Data availability

All data and materials generated or analyzed during this study are included in this manuscript (Figures and supplementary information).

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
