## [Editor Report]

The pathogenesis of human IgG4-mediated autoimmune diseases and the relevance of IgG4 autoantibodies remain incompletely understood and this study addresses an interesting issue on the role of IgG4 autoantibodies on disease pathology. By using FcgR knockout and engineered Abs, authors nicely show that IgG effector function mediates opposite functions, protecting or exacerbating the pathophysiology, depending on diseases.

---

## [Decision Letter]

**Decision letter after peer review:**

[Editors’ note: the authors submitted for reconsideration following the decision after peer review. What follows is the decision letter after the first round of review.]

Thank you for submitting your work entitled "Distinct impact of IgG subclass on autoantibody pathogenicity in different IgG4-mediated diseases" for consideration by *eLife*. Your article has been reviewed by 2 peer reviewers, and the evaluation has been overseen by a Reviewing Editor and a Senior Editor. The following individuals involved in review of your submission have agreed to reveal their identity: Falk Nimmerjahn (Reviewer #1).

We are sorry to say that, after consultation with the reviewers, we have decided that your work will not be considered further for publication by *eLife* as we estimated that too much revision experiments are needed to improve the study. However, the paper can be reconsidered as a new submission in the future if you are able to adequately address the issues raised by the reviewers.

Summary

The authors investigated the contribution of IgG4 subclass in the reduction or exarcerbation of autoantibody pathogenicity in IgG4-mediated autoimmune diseases, such as thrombotic thrombocytopenic purpura and pemphigus foliaceus. They notably show that the IgG4 subclass has a detrimental effect in thrombotic thrombocytopenic purpura while a beneficial effect in pemphigus foliaceus. While interesting, the reviewers expressed a major concern on the experimental design of the study, especially the murine adult pemphigus model in which the authors assessed the pathogenic activities of anti-Dsg1 antibodies that did not allowed to evaluate the blister formation per se. The reviewers also suggested to include Fab fragments in the pemphigus model to evaluate the difference in IgG subclass as IgG Fab from pemphigus sera is sufficient to induce blisters in mice. Please see below the reviewer's comments:

*Reviewer #1 (Recommendations for the authors (required)):*

The manuscript by Bi and colleagues addresses an interesting question. Understanding how different human IgG subclasses are contributing to tissue pathology is of great importance to allow predicting if patients with these autoantibodies are more prone to develop severe disease. One special issue represent human IgG 4 autoantibodies, as they on the one hand have a low capacity to bind Fc-receptors and complement, suggesting a low pro-inflammatory activity, but on the other hand are associated with major pathology in a variety of IgG4 mediated diseases. To address this issue, the authors chose two human autoimmune diseases, thrombotic thrombocytopenic purpura (TTP), in which IgG1 and IgG4 autoantibodies specific for ADAMTS13 are prevalent, and a skin disease called pemphigus foliaceus (PF) in which autoantibodies specific for Dsg1 cause skin blistering. By using different patient derived IgG1 and IgG4 autoantibodies and class switch variants thereof in wildtype, FcR-knockout or FcR humanized mice the authors demonstrate that in TTP both, IgG1 and IgG4 autoantibodies are pathogenic and cause ADMATS13 reduction. IgG1 dependent reduction of protease activity seemed to be longer lasting, however, which was at least in part dependent on Fc-receptors. With respect to Dsg1 specific autoantibodies both, IgG1 and IgG4 antibodies were causing skin inflammation. Unexpectedly, IgG variants lacking FcR binding, such as the N297A variant, or wildtype antibodies injected into FcR deficient mice were more pathogenic, suggesting that the autoantibody FcR interaction was limiting inflammation. Moreover, co-injection of antibodies with functional Fc-domains was reducing the pathogenic activity of autoantibodies with FcR-silent Fc-domains.

In summary, this is a very interesting study with great impact for the human clinical situation. There are some aspects that may need to be considered to further strengthen the impact of the study.

Points to be considered:

In Figure 2 it seems that the GASDALIE variant works less well than the non-mutated IgG1 variant. However, these two variants are never compared side by side. If so, this needs to be explained in more detail. In addition to the ELISA data in Figure S1 have the binding affinities been determined? It seems that most enhanced binding is to FcgRIV. Does this receptor play a role? Determining which activating mouse FcgR is critical for ADAMTS13 specific autoantibody activities would help to assess this a little better. Alternatively, the authors could use the human FcRtg mice instead of wildtype mice as they have done in Figure 1. This would allow the best comparison between the IgG1 wildtype and the GASDALIE variant.

Compared to the PF autoantibodies most of the activity of ADAMTS13 specific autoantibodies seems to be FcR independent. How do the authors explain that only the late recovery phase is dependent on FcRs? Why should this phase suddenly be dependent on FcRs?

In the PF in vivo experiments a critical piece of data that seems to be missing is data on the presence of the different IgG switch (IgG1, IgG4) and mutated (N297A, GASDALIE) in the skin as shown only as an example in Figure S4A. If GASDALIE variants never get to the tissue and are taken up by other FcR expressing cells this may be an alternative explanation here. Also data on the activation of complement proteins in the tissue may be of great value.

*Reviewer #2 (Recommendations for the authors (required)):*

In the previous studies, a variety of subclasses of autoantibodies have been serologically detected in autoantibody-mediated diseases. However, their definite biological function and impact on disease pathogenicity are not fully understood. Fc-FcγR interactions play an important role in humoral immune responses and their immunological functions are experimentally well elucidated. However, their biological behaviors in the pathophysiology of antibody-mediated autoimmune diseases are not fully clarified.

This paper showed the possibility that subclasses of IgG or Fc-FcγR interactions can influence the pathogenicity in thrombotic thrombocytopenic purpura (TTP) and pemphigus foliaceus (PF), two important autoantibody-mediated autoimmune diseases. This paper could be significant and bring new insight for research of the association of IgG subclasses and Fc-FcγR interactions with the pathophysiology of autoimmune diseases.

However, their experimental design does not necessarily properly address their questions on their pathogenic effects in pemphigus foliaceus. It is uncertain how much their observations could be applied to autoimmune diseases in general.

Strength

The strength of this study is their unique approach to explore the role of IgG subclass in the pathogenic activities and their attempt to compare them in mice models of two different autoimmune diseases. This study suggested the possibility that IgG4 subclass and Fc-FcγR interaction could influence the pathogenicity of the two antibody-mediated autoimmune diseases, TTP and PF. The results described here may expand our knowledge of the pathophysiology of these diseases, and the association of IgG subclasses and Fc-FcγR interactions with the pathogenicity of autoantibody-mediated autoimmune diseases.

Weakness

The weakness of this study is the relevance of their findings with mice model to support their claims. The authors should have included Fab fragments without Fc in comparison to IgG1 and IgG4 in all experiments to draw appropriate conclusion. In addition, the current experimental design results in evaluating the healing process or wound healing of superficial erosion rather than the blister formation in pemphigus. The FcγR-mediated clearance of apoptotic cells and immune complexes was not directly shown in their experiments. Therefore, the experimental design are their findings are not suitable and sufficient to support their conclusions.

1. The authors should have included Fab fragments without Fc as an important control in comparison to IgG1 and IgG4 in all experiments.

2. The authors studied pemphigus foliaceus, but this should be tested also in pemphigus vulgaris, which is the major subtype of pemphigus.

3. In pemphigus model, the authors used adult mice, which should represent chronic nature of pemphigus, rather than neonatal mice as an established model in pemphigus. However, they observed the mice only 3 to 4 days after injection, which should represent only short-term effects, and not reflect the recent description of lymphocyte infiltration at the skin lesions in patients. They evaluated the thickened epidermis and leukocyte infiltration which reflects would healing rather than blister formation. The authors do not necessarily evaluate the primary pathogenic effects of IgG autoantibodies injected. In particularly, In Figure 4D and E, Figure 5D and E, the authors analyzed the number and percentage of infiltrating neutrophils. Neutrophil infiltration is observed in the secondary inflammation phase after the onset of PF. In this phase, many factors such as secondary infection, scratching and other external stimulations can influence skin conditions. These data do not reflect the genuine pathogenicity of PF and FcγR mediated consequences.

4. The effects of anti-Dsg1 IgG are not evaluated in a quantitative fashion. The authors showed representative images of histology (Figure 4C, Figure 5A, Figure 7C, Figure S6A and B, Figure S7A), which is not sufficient to support the authors claims. More quantitative measurements such as scoring system of histological changes should be used.

5. Relating to comment #3, gross phenotype also should be evaluated in a quantitative fashion by introducing some scoring system.

6. The important question is whether the difference of pathogenicity between IgG1 and IgG4 autoantibodies is actually associated with the difference of affinity to FcγRs between them. Direct analysis to answer this question was not performed in the study. The authors could address the issue by determining whether the pathogenicity of IgG1 and IgG4 autoantibodies is affected in FcγR null mice.

7. The authors claim that IgG effector function can contribute to the FcγR-mediated clearance of autoantibody-Ag complexes and dead keratinocytes in this study. However, the FcγR-mediated clearance was not directly shown in the study. In Figure 6C, the analysis of Dsg1-Ab immune complex (IC) was conducted only once, and sequential evaluation was not performed. The current findings do not confirm that IC was actually cleared by FcγR-mediated effector function. The IC itself could be accelerated or attenuated in the early stages of the event. In Figure 6D, the author also analyzed the dead keratinocytes only once, and sequential data were also not shown.

8. In Figure 1B and D, after the administration of IgG1 and IgG4 antibodies, ADAMTS 13 activities were decreased similarly at first. Afterwards, ADAMTS 13 activities were recovered differently in IgG1 and IgG4 groups. It may be possible that this difference of recovery is not due to the difference of direct pathogenicity between these autoantibodies, but due to the difference of half-time of each antibody.

[Editors’ note: further revisions were suggested prior to acceptance, as described below.]

Thank you for submitting your article "Distinct impact of IgG subclass on autoantibody pathogenicity in different IgG4-mediated diseases" for consideration by *eLife*. Your article has been reviewed by 2 peer reviewers, and the evaluation has been overseen by a Reviewing Editor and Tadatsugu Taniguchi as the Senior Editor. The following individuals involved in review of your submission have agreed to reveal their identity: Falk Nimmerjahn (Reviewer #1).

Essential revisions:

Although both reviewers are positive for your manuscript, one reviewer has still some concerns. Thus, we encourage you to submit the revised manuscript, according to the following essential requests.

1) In figure 2, B and C, it has been showed that TTP1-420(IgG1)-treated WT and FcγRαnull mice displayed similar trend in the reduction of ADAMTS13 activity at day 1 and day 5 and at day 8 the level remains same for WT mice while FcγRαnull mice showed recovery, almost equivalent to control groups. At this time point, it can't be appropriately inferred if this recovery remains consistent further or changes after 2-4 days? Did the authors tested the ADAMTS13 activity in TTP1-420(IgG1) and TTP1-420(IgG4) treated WT, FcγRαnull and Fcer1g-/- mice beyond day 8? Did the TTP1-420(IgG1)-treated WT mice showed no recovery at all even on later time point? In order to draw any conclusion, authors should observe the ADAMTS13 activity for few more than 8 days.

2) How did authors claimed that the T cells doesn't have any role in the attenuation of the pathogenicity of anti-Dsg1 autoantibodies by Fc-FcγR interaction (Figure 5; Lines 348-350)? Direct experimental evidence in support of this hypothesis is lacking. Please provide the supporting data. Did they also examine any correlation between the T-cells subsets and disease pathogenesis? Hence, please change this sentence to avoid overstatement.

3) Authors observed the PF mice model for a very short span (only 2-4 days) after injecting the autoantibodies. What was the reason behind choosing the time-point for study? Increased infiltration of neutrophils (Figure 4D-F, Figure S5, A-C, Figure 5, D-E, Figure S6, E-G) was reported in the ears of FcγRαnull and hFCGRTg mice treated with different autoantibodies. However, analyzing neutrophils at such an early time-point doesn't give much conclusive information about the pathogenicity. Did they look for these parameters at a later stage too?

Although authors addressed many of the concerns raised by the previous reviewers, there remains some ambiguities in the experimental settings.

*Reviewer #2 (Recommendations for the authors):*

In the manuscript, Bi et al. studied the effects of IgG subclasses and Fc-FcγR interactions on autoantibody pathogenicity in human autoimmune diseases; thrombotic thrombocytopenic purpura (TTP) and pemphigus foliaceus (PF). The pathogenesis of IgG4 mediated autoimmune diseases and the relevance of IgG4 autoantibodies remain incompletely understood and this study addresses an interesting investigation on the role of IgG4 autoantibodies on disease pathology. The authors demonstrated protective mechanism of IgG4 and Fc-FcγR interaction in the pathogenicity of anti-ADAMTS13 autoantibodies isolated from TTP patients, whereas in case of anti-Dsg1 autoantibodies which causes skin blistering in PF, authors observed opposite effect, the FcγR mediated effector function attenuated the skin lesions, exacerbating the pathophysiology. Overall, the manuscript is largely well-written, however, enthusiasm is diminished. Although the present study offers a distinctive perspective in the understanding of the modes of action of IgG4 autoantibodies and disease pathogenesis, there are some ambiguities in the experimental set-up, and it also lacks some important information about human TTP samples and a proper interpretation of the study findings.

1) In figure 2, B and C, it has been showed that TTP1-420(IgG1)-treated WT and FcγRαnull mice displayed similar trend in the reduction of ADAMTS13 activity at day 1 and day 5 and at day 8 the level remains same for WT mice while FcγRαnull mice showed recovery, almost equivalent to control groups. At this time point, it can't be appropriately inferred if this recovery remains consistent further or changes after 2-4 days? Did the authors tested the ADAMTS13 activity in TTP1-420(IgG1) and TTP1-420(IgG4) treated WT, FcγRαnull and Fcer1g-/- mice beyond day 8? Did the TTP1-420(IgG1)-treated WT mice showed no recovery at all even on later time point? In order to draw any conclusion, authors should observe the ADAMTS13 activity for few more than 8 days.

2) Authors studied the effect of IgG4 autoantibodies on the ADAMTS13 antigen and activity in acquired TTP patients (Figure 3). Did they analyze clinical remission samples too and how does authors relate their findings in those patient samples? It has been reported in several articles that patients with acquired TTP who achieve clinical remission continue to show a persistent severe deficiency in ADAMTS13 activity(https://doi.org/10.1161/ATVBAHA.107.145797, https://doi.org/10.1182/blood-2016-06-724161, https://doi.org/10.3324/haematol.11739) and monitoring the association between IgG4 autoantibodies and ADAMTS13 activity in the remission samples may improve understanding of the clinical management of TTP patients.

3) How did authors claimed that the T cells doesn't have any role in the attenuation of the pathogenicity of anti-Dsg1 autoantibodies by Fc-FcγR interaction (Figure 5; Lines 348-350)? Direct experimental evidence in support of this hypothesis is lacking. Please provide the supporting data. Did they also examine any correlation between the T-cells subsets and disease pathogenesis?

4) Authors observed the PF mice model for a very short span (only 2-4 days) after injecting the autoantibodies. What was the reason behind choosing the time-point for study? Increased infiltration of neutrophils (Figure 4D-F, Figure S5, A-C, Figure 5, D-E, Figure S6, E-G) was reported in the ears of FcγRαnull and hFCGRTg mice treated with different autoantibodies. However, analyzing neutrophils at such an early time-point doesn't give much conclusive information about the pathogenicity. Did they look for these parameters at a later stage too?

As stated above, I agree with the previous reviewers that the experiments do not address the questions the manuscript purportedly addresses.

---

## [Author Response]

[Editors’ note: the authors resubmitted a revised version of the paper for consideration. What follows is the authors’ response to the first round of review.]

SummaryThe authors investigated the contribution of IgG4 subclass in the reduction or exarcerbation of autoantibody pathogenicity in IgG4-mediated autoimmune diseases, such as thrombotic thrombocytopenic purpura and pemphigus foliaceus. They notably show that the IgG4 subclass has a detrimental effect in thrombotic thrombocytopenic purpura while a beneficial effect in pemphigus foliaceus. While interesting, the reviewers expressed a major concern on the experimental design of the study, especially the murine adult pemphigus model in which the authors assessed the pathogenic activities of anti-Dsg1 antibodies that did not allowed to evaluate the blister formation per se. The reviewers also suggested to include Fab fragments in the pemphigus model to evaluate the difference in IgG subclass as IgG Fab from pemphigus sera is sufficient to induce blisters in mice. Please see below the reviewer's comments:Reviewer #1 (Recommendations for the authors (required)):The manuscript by Bi and colleagues addresses an interesting question. Understanding how different human IgG subclasses are contributing to tissue pathology is of great importance to allow predicting if patients with these autoantibodies are more prone to develop severe disease. One special issue represent human IgG 4 autoantibodies, as they on the one hand have a low capacity to bind Fc-receptors and complement, suggesting a low pro-inflammatory activity, but on the other hand are associated with major pathology in a variety of IgG4 mediated diseases. To address this issue, the authors chose two human autoimmune diseases, thrombotic thrombocytopenic purpura (TTP), in which IgG1 and IgG4 autoantibodies specific for ADAMTS13 are prevalent, and a skin disease called pemphigus foliaceus (PF) in which autoantibodies specific for Dsg1 cause skin blistering. By using different patient derived IgG1 and IgG4 autoantibodies and class switch variants thereof in wildtype, FcR-knockout or FcR humanized mice the authors demonstrate that in TTP both, IgG1 and IgG4 autoantibodies are pathogenic and cause ADMATS13 reduction. IgG1 dependent reduction of protease activity seemed to be longer lasting, however, which was at least in part dependent on Fc-receptors. With respect to Dsg1 specific autoantibodies both, IgG1 and IgG4 antibodies were causing skin inflammation. Unexpectedly, IgG variants lacking FcR binding, such as the N297A variant, or wildtype antibodies injected into FcR deficient mice were more pathogenic, suggesting that the autoantibody FcR interaction was limiting inflammation. Moreover, co-injection of antibodies with functional Fc-domains was reducing the pathogenic activity of autoantibodies with FcR-silent Fc-domains.In summary, this is a very interesting study with great impact for the human clinical situation. There are some aspects that may need to be considered to further strengthen the impact of the study.Points to be considered:In Figure 2 it seems that the GASDALIE variant works less well than the non-mutated IgG1 variant. However, these two variants are never compared side by side. If so, this needs to be explained in more detail. In addition to the ELISA data in Figure S1 have the binding affinities been determined? It seems that most enhanced binding is to FcgRIV. Does this receptor play a role? Determining which activating mouse FcgR is critical for ADAMTS13 specific autoantibody activities would help to assess this a little better. Alternatively, the authors could use the human FcRtg mice instead of wildtype mice as they have done in Figure 1. This would allow the best comparison between the IgG1 wildtype and the GASDALIE variant.

We thank the reviewer for the suggestion to compare hIgG1 and GASDALIE antiADAMTS13 antibodies in hFCGR^Tg^ mice directly. We agree with the reviewer that determining the binding affinity of the IgG variants used in the study to mouse FcγRs helps understand the FcγRs involved in anti-ADAMTS13 pathogenicity. To address these points, the following two additional experiments were performed as suggested by the reviewer:

i) The effect of TTP1-420(IgG1) and TTP1-420(GASDALIE) to inhibit the enzymatic activity of ADAMTS13 was directly compared in hFCGR^Tg^ mice. (line 194-199 in the updated manuscript):

“The pathogenicity of non-mutated IgG1 and GASDALIE variant was also compared side by side in hFCGR^Tg^ mice. Consistently, both TTP1-420(IgG1) and TTP1420(GASDALIE)-treated hFCGR^Tg^ mice induced a significant reduction in ADAMTS13 activity soon after the treatment (Figure 2H), and the recovery of ADAMTS13 activity is faster in TTP1-420(IgG1)-treated mice than in TTP1-420(GASDALIE)-treated mice (day 1 and day 5) (Figure 2, H and I).”

ii) SPR was performed to determine the binding affinity of the IgG variants used in the study to mouse FcγRs. Consistent with the ELISA results (Supplemental Figure 1B), the GASDALIE variant has a large increase in binding to mouse FcγRIV and FcγRIII (new Supplemental Figure 1C and new Supplemental Table1), which is also consistent with data published by a recent study of hIgG1 and GASDALIE binding affinity to mouse FcγRs by biolayer inferometry (Gunn et al., 2021). Since FcγRIV is the higher binding affinity mouse FcγR for GASDALIE, it is likely to contribute to the observed difference between hIgG1 and GASDALIE in the background of mouse FcγRs. At the same time, FcγRIV is also the mouse FcγR with the most pronounced binding affinity difference between hIgG1 and hIgG4. Overall, these data suggest a potential role of mouse FcγRIII, and in particular FcγRIV, for hIgG1 and GASDALIE ADAMTS13-specific autoantibody activities.

In hFCGR^Tg^ mice, GASDALIE has significantly increased binding to human FcγRIIIA and FcγRIIA (Gunn et al., 2021; Smith, DiLillo, Bournazos, Li, and Ravetch, 2012), these two human FcγRs may contribute to the observed difference between hIgG1 and GASDALIE ADAMTS13-specific autoantibody activities.

Reference paper:

Compared to the PF autoantibodies most of the activity of ADAMTS13 specific autoantibodies seems to be FcR independent. How do the authors explain that only the late recovery phase is dependent on FcRs? Why should this phase suddenly be dependent on FcRs?

We think the initiation of anti-ADAMTS13 and anti-Dsg1 autoantibody-induced pathological responses is primarily based on their blocking activity, which is based on their Fab and does not require FcγRs. This is supported by the rapid inhibition of ADAMTS13 activities by all the anti-ADAMTS13 variants we tested, regardless of their FcγR-binding properties, in various mouse strains we tested irrespective of their FcγR expression profiles, This is also supported by the pathogenicity of Fab, human IgG1, and IgG4 anti-Dsg1 antibodies in neonatal mice (new Supplemental Figure 3, B and C), and high dose of human IgG1 and IgG4 anti-Dsg1 antibodies in adult mice (Supplemental Figure 4). This is consistent with previous studies showing that pathogenic ADAMTS13specific autoantibodies could inhibit the activity of ADAMTS13 through blocking the active center of this enzyme, and Dsg1-specific autoantibodies could destroy the desmosome connections to induce skin blisters (Huijbers, Plomp, van der Maarel, and Verschuuren, 2018).

Our data support a model where the recovery phase of anti-ADAMTS13 and anti-Dsg1 autoantibody-induced pathological responses are both affected by FcγRs. After the onset of the pathological response, AutoAg-AutoAb immune complex will appear and accumulate. In the case of anti-ADAMTS13, FcγR-mediated clearance of AutoAg-AutoAb immune complexes will slow down the recovery, whereas in the case of anti-Dsg1, FcγRmediated clearance of apoptotic keratinocytes will promote the wound healing. The different effect during the recovery phase is due to the different impact of FcγR-mediated effector function on these different pathogenic responses.

Since the Fab-driven pathological response is more direct, and FcγR-mediated effect is also dependent on Fab, we think that despite that Fab-dependent effect and Fc-dependent effect coexist throughout the process, the initiation phase is dominated by Fab-driven pathological responses, and only in the recovery phase the impact of Fc-FcγR-mediated effector function was observed in both TTP and PF models.

In the PF in vivo experiments a critical piece of data that seems to be missing is data on the presence of the different IgG switch (IgG1, IgG4) and mutated (N297A, GASDALIE) in the skin as shown only as an example in Figure S4A. If GASDALIE variants never get to the tissue and are taken up by other FcR expressing cells this may be an alternative explanation here. Also data on the activation of complement proteins in the tissue may be of great value.

We agree with the reviewer that it is important to evaluate the accessibility of anti-Dsg1 antibodies to the skin tissues. As suggested, we analyzed the presence of human IgG antiDsg1 antibodies by immunofluorescence 6 and 24 hours after treatment, and confirmed that these antibodies (IgG1 and IgG4; N297A and GASDALIE) are present in the skin tissue at comparable levels (new Figure S4D and new Figure S8C). (line 301-303 and line 378-382 in the updated manuscript):

We also agree that detecting the activation of complement proteins is also helpful since IgG1 and IgG4 have different potency in activating complements. Previously, it has been reported that complement fixation was observed in the skin lesions of bullous pemphigoid patients (Hammers and Stanley, 2016). Previous studies have shown that the blisters formation in pemphigus is independent of complement using C5-deficient mice or blocking complement with cobra venom factor (Anhalt et al., 1986; Dainichi, Chow, and Kabashima, 2017). In our study, since IgG1 and IgG4 anti-Dsg1 antibodies displayed comparable pathogenicity in FcγR-null mice (Figure 4C, D), we think that complement is not likely to have a major impact, although an addictive role in FcγR-sufficient background cannot be ruled out.

Reviewer #2 (Recommendations for the authors (required)):In the previous studies, a variety of subclasses of autoantibodies have been serologically detected in autoantibody-mediated diseases. However, their definite biological function and impact on disease pathogenicity are not fully understood. Fc-FcγR interactions play an important role in humoral immune responses and their immunological functions are experimentally well elucidated. However, their biological behaviors in the pathophysiology of antibody-mediated autoimmune diseases are not fully clarified.This paper showed the possibility that subclasses of IgG or Fc-FcγR interactions can influence the pathogenicity in thrombotic thrombocytopenic purpura (TTP) and pemphigus foliaceus (PF), two important autoantibody-mediated autoimmune diseases. This paper could be significant and bring new insight for research of the association of IgG subclasses and Fc-FcγR interactions with the pathophysiology of autoimmune diseases.However, their experimental design does not necessarily properly address their questions on their pathogenic effects in pemphigus foliaceus. It is uncertain how much their observations could be applied to autoimmune diseases in general.StrengthThe strength of this study is their unique approach to explore the role of IgG subclass in the pathogenic activities and their attempt to compare them in mice models of two different autoimmune diseases. This study suggested the possibility that IgG4 subclass and Fc-FcγR interaction could influence the pathogenicity of the two antibody-mediated autoimmune diseases, TTP and PF. The results described here may expand our knowledge of the pathophysiology of these diseases, and the association of IgG subclasses and Fc-FcγR interactions with the pathogenicity of autoantibody-mediated autoimmune diseases.WeaknessThe weakness of this study is the relevance of their findings with mice model to support their claims. The authors should have included Fab fragments without Fc in comparison to IgG1 and IgG4 in all experiments to draw appropriate conclusion. In addition, the current experimental design results in evaluating the healing process or wound healing of superficial erosion rather than the blister formation in pemphigus. The FcγR-mediated clearance of apoptotic cells and immune complexes was not directly shown in their experiments. Therefore, the experimental design are their findings are not suitable and sufficient to support their conclusions.1. The authors should have included Fab fragments without Fc as an important control in comparison to IgG1 and IgG4 in all experiments.

Previously, it has been well established that Fab fragments of anti-Dsg1 antibodies without Fc is pathogenic in both neonatal mouse and human skin models, suggesting that Fc-FcγR interactions is not absolutely required for the pathogenicity of anti-Dsg1 antibodies. To address the point raised by the reviewer, we now included data confirming that in the neonatal model that Fab anti-Dsg1 antibodies are sufficient for inducing skin lesions (New Supplemental Figure 3, B and C), which is consistent with previous studies (Rock, Labib, and Diaz, 1990).

At the same time, we disagree that Fab fragments without Fc need to be added in all experiments studying IgG1 and IgG4 for the following reasons:

2. The authors studied pemphigus foliaceus, but this should be tested also in pemphigus vulgaris, which is the major subtype of pemphigus.

We agree with the reviewer that pemphigus vulgaris, which is induced by both anti-Dsg1 and Dsg3 autoantibodies, is more common than anti-Dsg1-induced pemphigus foliaceus, and further study of pemphigus vulgaris and other IgG4-mediated autoimmune diseases will help the understanding of these autoimmune diseases, which represent a major category of autoimmune diseases. However, as the first study of its kinds, we believe that our study of TTP and PF, two IgG4-mediated autoimmune diseases targeting different tissues, provides new insight into the understanding of these diseases, and further investigation of other IgG4-mediated autoimmune diseases using similar approaches is warranted but beyond the scope of the current study.

3. In pemphigus model, the authors used adult mice, which should represent chronic nature of pemphigus, rather than neonatal mice as an established model in pemphigus. However, they observed the mice only 3 to 4 days after injection, which should represent only short-term effects, and not reflect the recent description of lymphocyte infiltration at the skin lesions in patients. They evaluated the thickened epidermis and leukocyte infiltration which reflects would healing rather than blister formation. The authors do not necessarily evaluate the primary pathogenic effects of IgG autoantibodies injected. In particularly, In Figure 4D and E, Figure 5D and E, the authors analyzed the number and percentage of infiltrating neutrophils. Neutrophil infiltration is observed in the secondary inflammation phase after the onset of PF. In this phase, many factors such as secondary infection, scratching and other external stimulations can influence skin conditions. These data do not reflect the genuine pathogenicity of PF and FcγR mediated consequences.

The pathogenesis of PF involves both a process leading to autoantibody production and a pathogenic response after autoantibodies are produced (including the onset of blistering and the resolving of skin lesions). We agreed with the reviewer that our adult mouse model has limitations, which does not model the process leading to autoantibody production, including the development and differentiation of autoreactive B lymphocytes, such as the recent description of lymphocyte infiltration at the skin lesions in patients the reviewer mentioned (Yuan et al., 2017; Zhou et al., 2019). At the same time, our model does provide a new opportunity, which is not offered by the previously established neonatal mouse and the human skin models, to study a complete pathological response once the autoantibodies are provided, including both the onset of blistering and the resolving of skin lesions. This response lasts for a few days and occurs in vivo in the context of the complex environment of adult individuals. Therefore, our model is particularly useful for the studies not allowed by the neonatal mouse and the human skin models, which have been well-established but are limited to a few hours or in vitro study. To clarify the applications allowed by our model, we have revised the description of our animal model as an adult mouse model that allows for the study of a complete pathological response once the autoantibodies are provided, including both the onset of blistering and the resolving of skin lesion. (line 282-284 in the revised manuscript).

We also agree that it is essential to model the blister formation in our model. In fact, skin blistering was evaluated as a key feature in our model. As shown in Figures 5F and 7B, and in the new supplemental Figures 4A and 6I of the updated manuscript, skin blistering formation was visibly clear in our model after the anti-Dsg1 autoantibodies were injected. Furthermore, the histological examination also shows clear skin blisters formed at the skin lesions induced by anti-Dsg1 antibodies in our model (Figure 4C, 5A, 6E, 7C and supplemental Figure 3B, 4A, 4C, 6A, 6B, 7A) in addition to the thickened epidermis and leukocyte infiltration, which represents the downstream events following skin blistering. To quantify the severity of skin blistering, a blistering score system was used in the updated manuscript (an easy grading system similar to PDAI (Pemphigus Disease Area Index) according to the area of skin lesions.).

At the same time, we would like to point out that skin blistering, thickened epidermis, leukocyte infiltration (including neutrophil), and wound healing are all part of the pathological response induced by anti-Dsg1 autoantibodies in vivo. It is meaningful to evaluate these features in vivo for the understanding of PF pathogenesis. In fact, we think that being able to evaluate these features is the strength of our adult mouse model.

We also agree that neutrophil infiltration is observed in the secondary inflammation phase after the onset of PF, and many factors can influence skin conditions, including the factors named by the reviewer (such as secondary infection, scratching, and other external stimulations). However, our experiments were well controlled, so the variable is limited to the antibodies injected or the mouse strains used. In our study, we typically used 3~5 mice that are evenly distributed in different cages for each group, so the other factors could be better controlled. Indeed, the fact that neutrophil infiltration is observed in our adult mouse model demonstrates the relevance of the model as neutrophil infiltration has been reported in the skin lesion of PF patients (Furtado, 1959; Rados, 2011).

Therefore, while the previously established neonatal mouse model has been proven valuable to demonstrate anti-Dsg1-induced blister formation, our adult mouse model provides us an opportunity to evaluate both anti-Dsg1-induced blister formation and the downstream events.

4. The effects of anti-Dsg1 IgG are not evaluated in a quantitative fashion. The authors showed representative images of histology (Figure 4C, Figure 5A, Figure 7C, Figure S6A and B, Figure S7A), which is not sufficient to support the authors claims. More quantitative measurements such as scoring system of histological changes should be used.

As suggested by the reviewer, a blister-scoring system of histological changes was used to evaluate the severity of skin blistering quantitatively. Histological data were graded semi-quantitatively based on the percentage of the specimen with blisters (Mahoney, Wang, and Stanley, 1999), on the following scale: 0, no blister; 1, blisters at edge only, (< 25 %); 2, localized blisters, 25~50 % of the specimen; 3, extensive blisters, 50~75 % of the specimen; 4, very extensive blisters, > 75% of the specimen.

The histological change scores are summarized and provided in the updated manuscript.

Consistently, these results supported our conclusions.

5. Relating to comment #3, gross phenotype also should be evaluated in a quantitative fashion by introducing some scoring system.

We thank the reviewer for the suggestion. The gross phenotype is now evaluated in a quantitative fashion (score 0-4) according to the area of skin lesions. The data supports our conclusion.

6. The important question is whether the difference of pathogenicity between IgG1 and IgG4 autoantibodies is actually associated with the difference of affinity to FcγRs between them. Direct analysis to answer this question was not performed in the study. The authors could address the issue by determining whether the pathogenicity of IgG1 and IgG4 autoantibodies is affected in FcγR null mice.

We agree with the reviewer that “whether the difference of pathogenicity between IgG1 and IgG4 autoantibodies is actually associated with the difference of affinity to FcγRs between them” is an important question and needs to be clarified in the context of our findings. To address this question, we have performed the experiment suggested by the reviewer: to determine whether the pathogenicity of IgG1 and IgG4 autoantibodies is affected in FcγR null mice (FcγRα^null^). The results show that FcγR deficiency could exacerbate the pathogenicity of both anti-Dsg1 IgG1 and IgG4 (Figure 4, C-F). At the same time, no significant differences were noted between the severity of skin lesions induced by anti-Dsg1 IgG1 and IgG4 antibodies in FcγR null mice (Figure 4, C-F ), supporting the notion that the difference in the pathogenicity of IgG1 and IgG4 anti-Dsg1 antibodies depends on FcγR expression.

7. The authors claim that IgG effector function can contribute to the FcγR-mediated clearance of autoantibody-Ag complexes and dead keratinocytes in this study. However, the FcγR-mediated clearance was not directly shown in the study. In Figure 6C, the analysis of Dsg1-Ab immune complex (IC) was conducted only once, and sequential evaluation was not performed. The current findings do not confirm that IC was actually cleared by FcγR-mediated effector function. The IC itself could be accelerated or attenuated in the early stages of the event. In Figure 6D, the author also analyzed the dead keratinocytes only once, and sequential data were also not shown.

As the reviewer suggested, the sequential evaluation of Dsg1-Ab immune complex (IC) (new Supplemental Figure 8B) and dead keratinocytes (new Supplemental Figure 8D) was performed at 6 and 24 hours after autoantibody injections. In order to further test whether IgG effector function can contribute to the FcγR-mediated clearance of autoantibody-Ag complexes and dead keratinocytes, Dsg1-Ab immune complex (IC) and dead keratinocytes were also evaluated in FcγRα^null^ mice (new Figure 6, D to F). These studies showed that the levels of Dsg1-autoantigen-autoantibody immune complexes were clearly higher in PF1-8-15(N297A)-treated nude mice than in PF1-8-15(GASDALIE)treated nude mice, but not in FcγRα^null^ mice (Figure 6D, and Figure S8B). In addition, we detected comparable levels of N297A and GASDALIE anti-Dsg1 antibodies in the skin tissue of nude mice 6 and 24 hours after treatment (Figure S8C), suggesting the different pathogenic functions of these two antibodies are not due to different accessibility to the skin tissues. Notably, we observed more epidermal blistering and dead keratinocytes at skin lesions in PF1-8-15(N297A)-treated nude mice than in PF1-8-15(GASDALIE)treated nude mice (Figure 6, E and F, and Figure S8D). More epidermal blistering and dead keratinocytes were also observed in FcγRα^null^ mice regardless of whether PF1-815(N297A) or PF1-8-15(GASDALIE) was used (Figure 6, E and F). Our study of these anti-Dsg1 variants with different FcγR binding properties in mice with different FcγRbackground (FcγR-sufficient and deficient mice) supports the notion that FcγR-mediated effector function promotes the clearance of autoantibody-Ag complexes and dead keratinocytes. These results are discussed at line 371-387 in the updated manuscript.

8. In Figure 1B and D, after the administration of IgG1 and IgG4 antibodies, ADAMTS 13 activities were decreased similarly at first. Afterwards, ADAMTS 13 activities were recovered differently in IgG1 and IgG4 groups. It may be possible that this difference of recovery is not due to the difference of direct pathogenicity between these autoantibodies, but due to the difference of half-time of each antibody.

We thank the reviewer for raising this point. The half-life of IgG is determined by its binding affinity to FcRn, and it has been established that the half-life of both IgG1 and IgG4 is about 21 days (Vidarsson, Dekkers, and Rispens, 2014). More recently, a comparable half-life of human IgG1 and IgG4 was reported in mice and several other animal models (Tabrizi et al., 2019). Furthermore, our study of N297A and GASDALIE variants of anti-ADAMTS13 antibodies further support the notion that the observed difference between these variants is not due to different half-life since all mutations in these variants are located at positions that do not interact or affect FcRn binding (Burmeister, Huber, and Bjorkman, 1994; Shields et al., 2001).

Anhalt, G. J., Till, G. O., Diaz, L. A., Labib, R. S., Patel, H. P., and Eaglstein, N. F. (1986). Defining the role of complement in experimental pemphigus vulgaris in mice. J Immunol, 137(9), 2835-2840.

Bournazos, S., Klein, F., Pietzsch, J., Seaman, M. S., Nussenzweig, M. C., and Ravetch, J. V. (2014). Broadly neutralizing anti-HIV-1 antibodies require Fc effector functions for in vivo activity. Cell, 158(6), 1243-1253. doi:10.1016/j.cell.2014.08.023

Burmeister, W. P., Huber, A. H., and Bjorkman, P. J. (1994). Crystal structure of the complex of rat neonatal Fc receptor with Fc. Nature, 372(6504), 379-383. doi:10.1038/372379a0

Dainichi, T., Chow, Z., and Kabashima, K. (2017). IgG4, complement, and the mechanisms of blister formation in pemphigus and bullous pemphigoid. J Dermatol Sci, 88(3), 265-270. doi:10.1016/j.jdermsci.2017.07.012

Dekkers, G., Bentlage, A. E. H., Stegmann, T. C., Howie, H. L., Lissenberg-Thunnissen, S., Zimring, J.,... Vidarsson, G. (2017). Affinity of human IgG subclasses to mouse Fc γ receptors. MAbs, 9(5), 767-773. doi:10.1080/19420862.2017.1323159

Derebe, M. G., Nanjunda, R. K., Gilliland, G. L., Lacy, E. R., and Chiu, M. L. (2018). Human IgG subclass cross-species reactivity to mouse and cynomolgus monkey Fcgamma receptors. Immunol Lett, 197, 1-8. doi:10.1016/j.imlet.2018.02.006

Furtado, T. A. (1959). Histopathology of pemphigus foliaceus. AMA Arch Derm, 80(1), 66-71. doi:10.1001/archderm.1959.01560190068010

Gunn, B. M., Lu, R., Slein, M. D., Ilinykh, P. A., Huang, K., Atyeo, C.,... Alter, G. (2021). A Fc engineering approach to define functional humoral correlates of immunity against Ebola virus. Immunity, 54(4), 815-828 e815. doi:10.1016/j.immuni.2021.03.009

Hammers, C. M., and Stanley, J. R. (2016). Mechanisms of Disease: Pemphigus and Bullous Pemphigoid. Annu Rev Pathol, 11, 175-197. doi:10.1146/annurev-pathol012615-044313

Huijbers, M. G., Plomp, J. J., van der Maarel, S. M., and Verschuuren, J. J. (2018). IgG4mediated autoimmune diseases: a niche of antibody-mediated disorders. Ann N Y Acad Sci, 1413(1), 92-103. doi:10.1111/nyas.13561

Lux, A., Yu, X., Scanlan, C. N., and Nimmerjahn, F. (2013). Impact of immune complex size and glycosylation on IgG binding to human FcgammaRs. J Immunol, 190(8), 4315-4323. doi:10.4049/jimmunol.1200501

Mahoney, M. G., Wang, Z. H., and Stanley, J. R. (1999). Pemphigus vulgaris and pemphigus foliaceus antibodies are pathogenic in plasminogen activator knockout mice. J Invest Dermatol, 113(1), 22-25. doi:10.1046/j.1523-1747.1999.00632.x

Nimmerjahn, F., Bruhns, P., Horiuchi, K., and Ravetch, J. V. (2005). FcgammaRIV: a novel FcR with distinct IgG subclass specificity. Immunity, 23(1), 41-51.

doi:10.1016/j.immuni.2005.05.010

Rados, J. (2011). Autoimmune blistering diseases: histologic meaning. Clin Dermatol,

29(4), 377-388. doi:10.1016/j.clindermatol.2011.01.007

Rock, B., Labib, R. S., and Diaz, L. A. (1990). Monovalent Fab' immunoglobulin fragments from endemic pemphigus foliaceus autoantibodies reproduce the human disease in neonatal Balb/c mice. J Clin Invest, 85(1), 296-299. doi:10.1172/JCI114426

Shields, R. L., Namenuk, A. K., Hong, K., Meng, Y. G., Rae, J., Briggs, J.,... Presta, L. G. (2001). High resolution mapping of the binding site on human IgG1 for Fc γ RI, Fc γ RII, Fc γ RIII, and FcRn and design of IgG1 variants with improved binding to the Fc γ R. J Biol Chem, 276(9), 6591-6604. doi:10.1074/jbc.M009483200

Smith, P., DiLillo, D. J., Bournazos, S., Li, F., and Ravetch, J. V. (2012). Mouse model recapitulating human Fcgamma receptor structural and functional diversity. Proc Natl Acad Sci U S A, 109(16), 6181-6186. doi:10.1073/pnas.1203954109

Tabrizi, M., Neupane, D., Elie, S. E., Shankaran, H., Juan, V., Zhang, S.,... Escandon, E. (2019). Pharmacokinetic Properties of Humanized IgG1 and IgG4 Antibodies in Preclinical Species: Translational Evaluation. AAPS J, 21(3), 39. doi:10.1208/s12248-019-0304-3

Vidarsson, G., Dekkers, G., and Rispens, T. (2014). IgG subclasses and allotypes: from structure to effector functions. Front Immunol, 5, 520. doi:10.3389/fimmu.2014.00520

Yuan, H., Zhou, S., Liu, Z., Cong, W., Fei, X., Zeng, W.,... Pan, M. (2017). Pivotal Role of Lesional and Perilesional T/B Lymphocytes in Pemphigus Pathogenesis. J Invest Dermatol, 137(11), 2362-2370. doi:10.1016/j.jid.2017.05.032

Zhou, S., Liu, Z., Yuan, H., Zhao, X., Zou, Y., Zheng, J., and Pan, M. (2019). Autoreactive B Cell Differentiation in Diffuse Ectopic Lymphoid-Like Structures of Inflamed Pemphigus Lesions. J Invest Dermatol. doi:10.1016/j.jid.2019.07.717

[Editors’ note: what follows is the authors’ response to the second round of review.]

The reviewers have discussed their reviews with one another, and the Reviewing Editor has drafted this to help you prepare a revised submission.Essential revisions:Although both reviewers are positive for your manuscript, one reviewer has still some concerns. Thus, we encourage you to submit the revised manuscript, according to the following essential requests.

We thank both the reviewers and editors for the constructive comments. We have revised the manuscript according to the requests by the reviewers.

1) In figure 2, B and C, it has been showed that TTP1-420(IgG1)-treated WT and FcγRαnull mice displayed similar trend in the reduction of ADAMTS13 activity at day 1 and day 5 and at day 8 the level remains same for WT mice while FcγRαnull mice showed recovery, almost equivalent to control groups. At this time point, it can't be appropriately inferred if this recovery remains consistent further or changes after 2-4 days? Did the authors tested the ADAMTS13 activity in TTP1-420(IgG1) and TTP1-420(IgG4) treated WT, FcγRαnull and Fcer1g-/- mice beyond day 8? Did the TTP1-420(IgG1)-treated WT mice showed no recovery at all even on later time point? In order to draw any conclusion, authors should observe the ADAMTS13 activity for few more than 8 days.

We agree with the reviewer that it is important to find out whether the recovery we observed is transit or remains consistent further. To address this question, we analyzed the ADAMTS13 activity in TTP1-420(IgG1)-treated FcγRα-null mice, and observed that after the recovery of ADAMTS13 activity on day 8, there was no decrease in the following week (Author response image 1, data not shown in the revised manuscript).

**Author response image 1. sa2fig1:** The recovery of ADAMTS13 activity remains consistent for at least 1 week. We treated the FcγRα-null mice (n = 5) with 10 μg TTP1-420(IgG1) antibodies on day 0 through tail vein injection, blood was collected on day -1, day 1, day 6, day 8, day 11 and day 14 and analyzed for ADAMTS13 activity (A) and plasma remaining hIgG levels (B).

We also thank the reviewer for pointing out that it is important to find out whether TTP1-420(IgG1)-treated mice recover after day 8. As suggested by the reviewer, we added day 11 in our analysis of ADAMTS13 activity in TTP1-420(IgG1)-treated hFCGR-Tg mice. As shown in Figure R2 (Figure1—figure supplement 2 of the revised manuscript), ADAMTS13 activity in TTP1-420(IgG1)-treated hFCGR-Tg mice was reduced on days 1, 4, and 8, but recovered on day 11. We also observed that TTP1-420(IgG1) antibody levels in the plasma fell to the background levels (day -1) on day 11, which is not surprising given that only 10 μg of antibody was administered. These results suggest that ADAMTS13 activity can recover after the autoantibody is depleted, considering that this is a passive transfer model. Therefore, the model used in our study represents an acute model for evaluating the pathogenicity of anti-ADAMTS13 antibodies, where the best time window to analyze ADAMTS13 activity is before its recovery. This point is added to the manuscript at line 141-144:

“The ADAMTS13 activity in hFCGR-Tg mice treated with the same amount of TTP1-420(IgG1) was later recovered (Figure1—figure supplement 2, A-C), suggesting our model represents an acute model for evaluating the pathogenicity of anti-ADAMTS13 antibodies before the recovery of ADAMTS13 activity.”

2) How did authors claimed that the T cells doesn't have any role in the attenuation of the pathogenicity of anti-Dsg1 autoantibodies by Fc-FcγR interaction (Figure 5; Lines 348-350)? Direct experimental evidence in support of this hypothesis is lacking. Please provide the supporting data. Did they also examine any correlation between the T-cells subsets and disease pathogenesis? Hence, please change this sentence to avoid overstatement.

We thank the reviewer for pointing out that the statements about T cells need to be clarified. Since we observed that, in both T cell-sufficient hFCGR-Tg mice (Figure 5A) and T cell-deficient nude mice (Figure 5F), anti-Dsg1(N297A) autoantibodies induced more severe skin lesions as compared to matched anti-Dsg1(GASDALIE) autoantibodies, we think it is worth noting that “Fc-FcγR interaction attenuates the pathogenicity of anti-Dsg1 autoantibodies” and this is observed in both hFCGR-Tg and nude mice. We do not intend to claim that the T cells have no role in the process. To clarify this point, the statement was revised to “it is worth noting that this is observed in both hFCGR-Tg and nude mice” at line 318-319 in the revised manuscript.

3) Authors observed the PF mice model for a very short span (only 2-4 days) after injecting the autoantibodies. What was the reason behind choosing the time-point for study? Increased infiltration of neutrophils (Figure 4D-F, Figure S5, A-C, Figure 5, D-E, Figure S6, E-G) was reported in the ears of FcγRαnull and hFCGRTg mice treated with different autoantibodies. However, analyzing neutrophils at such an early time-point doesn't give much conclusive information about the pathogenicity. Did they look for these parameters at a later stage too?

We thank the reviewer for discussing the timing for analyzing anti-Dsg1 autoantibody-induced pathogenicity, which is an important aspect of the anti-Dsg1-induced PF model used in our study. As suggested by the reviewer, mice were studied for longer to understand the model better. WT mice treated with PF1-8-15(N297A) and PF1-8-15(GASDALIE) antibodies were observed to monitor the process of skin lesion onset and recovery. As shown in Figure R3 (Figure 5—figure supplement 2 of the revised manuscript), the process takes 2~3 weeks, after which the recovery was confirmed by both the histological and visual examinations (Figure 5—figure supplement 2, A and B), and by quantifying the levels of infiltrating neutrophils (Figure 5—figure supplement 2, C and D). The levels of serum human IgG dropped to undetectable on day 6 (Figure 5—figure supplement 2E), and can’t be detected on day 23 in the skin tissue by DIF (Figure 5—figure supplement 2F). These results suggest that the first few days seem the best time window for studying anti-Dsg1 pathogenicity. These results were added into the revised manuscript at line 304-312:

“Consisting results were also observed in WT mice treated with PF1-8-15(N297A) and PF1-8-15(GASDALIE) antibodies when the onset and recovery of skin lesions were monitored (Figure 5—figure supplement 2A). The process lasted 2~3 weeks, and the recovery was confirmed by the histological and vision examinations (Figure 5—figure supplement 2, A and B), and by quantifying the infiltrating neutrophils (Figure 5—figure supplement 2, C and D). At the same time, the levels of serum human IgG dropped to undetectable on day 6 (Figure 5—figure supplement 2E), and can’t be detected on day 23 in the skin tissue by DIF (Figure 5—figure supplement 2F). These results further support that the first few days seem the best time window for studying anti-Dsg1 pathogenicity.”

We also would like to point out that, since the most direct pathogenic impact of anti-Dsg1 autoantibodies is causing skin lesions rather than inflammation, the analysis of neutrophils was performed as another readout of skin lesions in addition to histological and visual examination of skin lesions. More specifically, the pathogenicity of anti-Dsg1 antibodies is triggered by intraepidermal superficial blistering, the lesion in the skin epidermis that can be examined by histological examination. Neutrophils were analyzed because they are very sensitive to lesions and are considered the first leukocyte subset arriving at lesion sites (de Oliveira, Rosowski, and Huttenlocher, 2016; Rados, 2011), which can be examined by HE staining or FACS analysis. Therefore, histological and visual examinations, together with neutrophil analysis, collectively provide information on anti-Dsg1 pathogenicity. In order to address these points, the following statement was added into the manuscript at line 282-284:

“…neutrophils that are very sensitive to lesions and considered the first leukocyte subset arriving at lesion sites (de Oliveira et al., 2016; Rados, 2011)”

In addition, a supplemental Figure showing the typical histological features of skin lesions was included in the revised manuscript (Figure 4-supplement 3B of the revised manuscript):

In order to address these points, the following statement was added into the manuscript at line 256-257:

“…as well as crusting and scaling during the recovery (Figure 4—figure supplement 3, A and B)”.

References

de Oliveira, S., Rosowski, E. E., and Huttenlocher, A. (2016). Neutrophil migration in infection and wound repair: going forward in reverse. Nat Rev Immunol, 16(6), 378-391. doi:10.1038/nri.2016.49

Rados, J. (2011). Autoimmune blistering diseases: histologic meaning. Clin Dermatol, 29(4), 377-388. doi:10.1016/j.clindermatol.2011.01.007

Sinkovits, G., Szilagyi, A., Farkas, P., Inotai, D., Szilvasi, A., Tordai, A.,... Prohaszka, Z. (2018). Concentration and Subclass Distribution of Anti-ADAMTS13 IgG Autoantibodies in Different Stages of Acquired Idiopathic Thrombotic Thrombocytopenic Purpura. Front Immunol, 9, 1646. doi:10.3389/fimmu.2018.01646